# Dynamic allostery in substrate binding by human thymidylate synthase

**Jeffrey P Bonin[1], Paul J Sapienza[2], Andrew L Lee[1,2]***

[1]Department of Biochemistry and Biophysics, School of Medicine, University of North Carolina, Chapel Hill, United States; [2]Division of Chemical Biology and Medicinal Chemistry, Eshelman School of Pharmacy, University of North Carolina, Chapel Hill, United States

**Abstract** Human thymidylate synthase (hTS) is essential for DNA replication and therefore a therapeutic target for cancer. Effective targeting requires knowledge of the mechanism(s) of regulation of this 72 kDa homodimeric enzyme. Here, we investigate the mechanism of binding cooperativity of the nucleotide substrate. We have employed exquisitely sensitive methyl-based CPMG and CEST NMR experiments enabling us to identify residues undergoing bifurcated linear 3-state exchange, including concerted switching between active and inactive conformations in the apo enzyme. The inactive state is populated to only ~1.3%, indicating that conformational selection contributes negligibly to the cooperativity. Instead, methyl rotation axis order parameters, determined by [2]H transverse relaxation rates, suggest that rigidification of the enzyme upon substrate binding is responsible for the entropically-driven cooperativity. Lack of the rigidification in product binding and substrate binding to an N-terminally truncated enzyme, both non-cooperative, support this idea. In addition, the lack of this rigidification in the N-terminal truncation indicates that interactions between the flexible N-terminus and the rest of the protein, which are perturbed by substrate binding, play a significant role in the cooperativity—a novel mechanism of dynamic allostery. Together, these findings yield a rare depth of insight into the substrate binding cooperativity of an essential enzyme.

**\*For correspondence:**
drewlee@unc.edu

**Competing interest:** The authors declare that no competing interests exist.

## Editor's evaluation

Bonin et al. provide important new insights into the nucleotide substrate binding cooperativity of homodimeric human thymidylate synthase (hTS). hTS converts deoxyuridine monophosphate (dUMP) into deoxythymidine monophosphate (dTMP) and is therefore a target for cancer therapies. Extensive use of methyl-based NMR spectroscopy provides a set of convincing data that support a number of new insights into the conformational preferences of the apo state and the role of conformational entropy in mediating cooperativity within this enzyme. In addition, the findings point to a key role for the intrinsically disordered N-terminal region of hTS in the dynamic changes that occur upon binding of the dUMP substrate. The work very nicely demonstrates the power of NMR spectroscopy in elucidating ensemble conformations and dynamics and creates new ways to think about targeting this essential enzyme.

## Introduction

Thymidylate synthases (TS) are enzymes which convert deoxyuridine monophosphate (dUMP) and methylenetetrahydrofolate (mTHF) into deoxythymidine monophosphate (TMP) and dihydrofolate (DHF), making them part of the only de novo pathway for production of the nucleotide thymidine. Disruption of this pathway causes cells to die a 'thymineless death'. Consequently, human TS (hTS)

is a therapeutic target for cancer. Effective targeting requires detailed knowledge of the mechanism of regulation of this 72 kDa homodimeric enzyme, which is currently lacking. TSs from several species, including human, are reported to display half-the-sites activity, meaning that only one of the two subunits can perform catalysis at a time (*Anderson et al., 1999*; *Johnson et al., 2002*; *Maley et al., 1995*). In addition, we have shown previously that the two subunits of hTS bind the nucleotide substrate dUMP with positive cooperativity (*Bonin et al., 2019*). These properties clearly demonstrate that intersubunit communication plays a major role in the regulation of the activity of hTS (i.e. these properties influence the enzymatic activity), but the physical mechanism(s) underlying this communication are not understood. Here, we focus on the mechanism of substrate binding cooperativity (difference in the free energies of the two dUMP binding events).

A potential clue regarding the mechanism of intersubunit communication lies in the presence of crystal structures of apo hTS in two distinct conformations of the catalytically essential active site loop (*Chen et al., 2017*; *Figure 1c*). It has been suggested that the apo enzyme exchanges between these so-called 'active' and 'inactive' conformations (*Phan et al., 2001*), and that this conformational equilibrium may drive the binding cooperativity via coupling to nucleotide binding (*Bonin et al., 2019*). This would necessitate that the inactive conformation be predominant in the apo form, which has been reported on the basis of fluorescence measurements (*Phan et al., 2001*). Further, the inactive conformation has been reported to be responsible for the interaction between hTS and its mRNA, which plays a role in regulation of the translation of hTS (*Brunn et al., 2014*) and is central to the mechanism of resistance to hTS-targeted chemotherapy (*Tai et al., 2004*). This putative conformational equilibrium is thus of interest for the regulation of hTS at multiple levels.

Another possible mechanism of communication centers around changes in the conformational entropy of the enzyme in response to substrate binding (i.e. a change in the dynamics of hTS as opposed to a change in the structure, termed 'dynamic allostery' *Cooper and Dryden, 1984*). Such a mechanism has been reported in other systems (*Capdevila et al., 2017*; *Petit et al., 2009*; *Popovych et al., 2006*; *Saavedra et al., 2018*), and would be consistent with our calorimetric measurements on hTS showing that the binding cooperativity is entropically driven (*Bonin et al., 2019*). Recent work from Wand and Sharp demonstrates that order parameters determined from nuclear magnetic resonance (NMR) relaxation measurements can be used to obtain reasonable quantifications of conformational entropy changes (*Caro et al., 2017*). As a result, it is now possible to compare NMR derived order parameter changes to changes in binding entropies determined calorimetrically, enabling us to assess the contribution of dynamic allostery to intersubunit communication observed in hTS. It is also possible that both of these mechanisms contribute to the substrate binding cooperativity.

In this work, we present an extensive characterization of the structure and dynamics of hTS in apo and dUMP bound forms using methyl-based NMR spectroscopy. A combined analysis of Carr-Purcell-Meiboom-Gill (CPMG) relaxation dispersion and chemical exchange saturation transfer (CEST) data show that apo hTS indeed undergoes a concerted exchange process, with some probes exchanging between three states, that is suppressed upon dUMP binding. There are also faster, localized motions involving exchange with "exposed states" which are still present in the dUMP bound form. We find, however, that apo hTS primarily adopts the active conformation, rather than the inactive as had been previously reported. As a result, the coupling of this exchange process to dUMP binding contributes negligibly to the binding cooperativity. Methyl rotation axis order parameters calculated from $^2$H transverse relaxation rates in apo and dUMP bound forms of hTS show an overall rigidification of the enzyme upon substrate binding corresponding to a change in conformational entropy, $-T\Delta S_{conf}$, of about 10 kcal/mol, enough to explain the roughly 5 kcal/mol difference between the two binding entropies if most of this rigidification occurs upon the first binding event. In addition, we find that hTS binding to the product TMP, as well as an N-terminal truncation of hTS binding to dUMP, lack both binding cooperativity and rigidification, further establishing the correlation between hTS dynamics and binding thermodynamics. This suggests that differences in the change in conformational entropy of the enzyme upon each of the two substrate binding events are responsible for the observed cooperativity. Finally, by creating a thermodynamic cycle using the changes in conformational entropy for full length and N-terminal truncated hTS binding to dUMP, we demonstrate that the intrinsically disordered 25 residue N-terminus makes a key contribution to the change in conformational entropy upon dUMP binding to the full length enzyme via interactions with the protein surface, a novel mechanism of dynamic allostery.

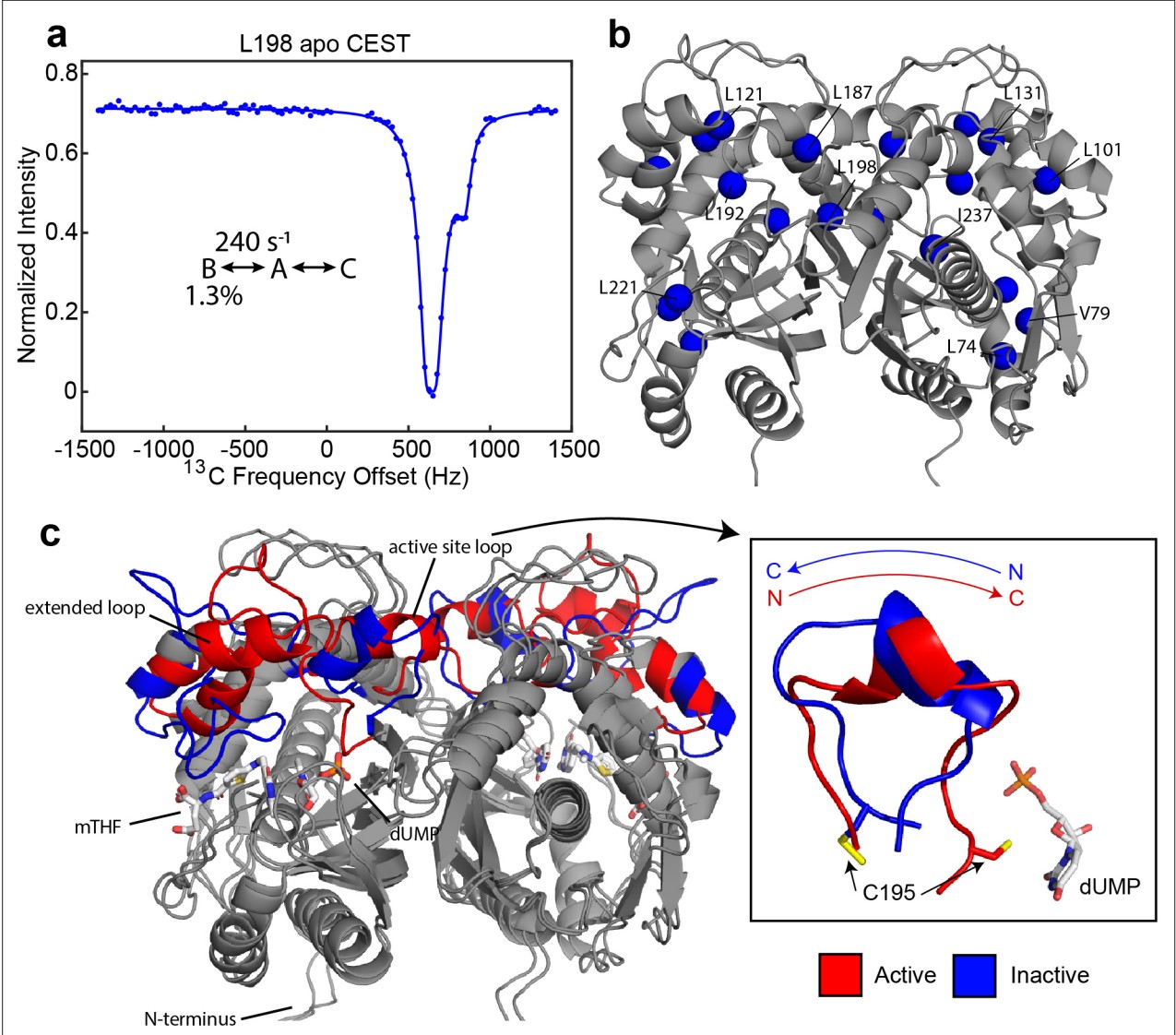

**Figure 1.** Apo hTS undergoes a slow, concerted motion which is likely exchange between active and inactive conformations. (**a**) $^{13}$C CEST profile of L198 in apo hTS at 25 Hz spin lock power (*Figure 1—source data 2*) with curve from the global fit. The CEST experiment is essentially only sensitive to the slower, concerted process. (**b**) Structure of apo hTS (active conformation, PDB ID 5X5A) showing the 10 probes involved in the slow, concerted motion. Many of these probes, including L101, L121, L131, L187, L192, and L198 are within or adjacent to the regions highlighted in (**c**) which show large changes between the active and inactive conformations. (**c**) Overlay of active (PDB ID 5X5A) and inactive (PDB ID 1YPV, residues 108–129 modelled in) structures of hTS. Substrates dUMP and mTHF are included in sticks. Red and blue colors highlight the areas that are most different between the two conformations, including the active site loop (residues 181–196) which undergoes a 180° flip (box on right) and the 'extended loop' (residues 100–135, most of which lack electron density in the inactive structure).

The online version of this article includes the following source data and figure supplement(s) for figure 1:

**Source data 1.** Apo hTS ILV $^{13}$C CEST 40 Hz spin lock.

**Source data 2.** Apo hTS ILV $^{13}$C CEST 25 Hz spin lock.

**Figure supplement 1.** CPMG and CEST profiles from concerted process probes.

## Results

### Apo hTS undergoes slower exchange between active and inactive conformations, in addition to faster localized processes on the µs-ms timescale

To understand the origin of binding cooperativity in hTS, we first needed to determine if a

conformational selection or dynamic allostery mechanism is utilized. In other words, does dUMP binding elicit a change in the structure or the dynamics of hTS, or perhaps both? To begin to answer this question, we needed to establish whether apo hTS, which crystallizes in active and inactive conformations (*Chen et al., 2017*; *Figure 1c*), switches between these two conformations in solution. NMR spectroscopy enables the investigation of protein dynamics on a wide range of timescales, including the µs-ms timescale where this type of conformational equilibrium is typically seen. For our characterization of the dynamics of hTS on the µs-ms timescale, we have employed CPMG relaxation dispersion and CEST experiments targeting the methyl groups on isoleucine, leucine, and valine side chains. These experiments probe somewhat overlapping regions of the µs-ms timescale, with CEST characterizing motions on the ms timescale and CPMG characterizing motions with exchange timescales of 100s of µs to 10s of ms. In the CEST experiment, a spin lock is applied at many frequencies scanning the entire spectrum, and the intensity of the observed (major state) signals are monitored to determine frequencies at which saturation is transferred from a minor state to the major state (*Vallurupalli et al., 2012*). In the CPMG experiment, dephasing of transverse magnetization due to stochastic exchange between states with distinct frequencies (i.e. the contribution of chemical exchange to transverse relaxation, $R_{ex}$) is refocused using trains of 180° pulses applied at various rates. We have used multiple-quantum (MQ) (*Korzhnev et al., 2004*) and $^{13}$C single-quantum (SQ) (*Lundström et al., 2007*) CPMG experiments as well as a $^{13}$C CEST experiment using the CHD$_2$ methyl isotopomer (*Rennella et al., 2015*). The MQ CPMG and CEST experiments are especially well suited to probing these motions in a large protein like hTS. Our analysis of the µs-ms dynamics of apo hTS include 850 and 600 MHz MQ CPMG datasets, an 850 MHz $^{13}$C SQ CPMG dataset, as well as 40 and 25 Hz spin lock power CEST datasets. For dUMP bound hTS, the analysis includes 850 and 600 MHz MQ CPMG datasets and an 850 MHz $^{13}$C SQ CPMG dataset.

The initial analysis of the CPMG and CEST datasets revealed that, while some methyl probes can be well-described as undergoing exchange between two conformations, other probes clearly cannot and appear to be involved in dynamic switching between three states (Appendix 1). Global analysis of the CPMG and CEST datasets (see Materials and methods) on apo hTS show that 10 probes are involved in a concerted process with an exchange rate of 240 ± 7 s$^{-1}$ and an excited state population of ~1.3% (*Figure 1*, *Figure 1—figure supplement 1*, *Supplementary file 2*, *Figure 1—source data 1 and 2*, *Figure 2—source data 1–4*). Interestingly, 8 of these 10 probes show 3-state exchange (Appendix 1), that in addition to the 2-state concerted process includes a faster process with variable exchange rate and population (*Figure 2*). Multiple trials of a global fit using a combination of 2- and 3-state models for different methyl sites were performed. For instances of 3-state behavior, a B↔A↔C model, in which each excited state (B and C) exchanges with the ground state (A) but not with each other, is most appropriate in these cases, as a 2-state model does not fit the data well and the A↔B↔C model lacks consistency in the exchange parameters amongst these probes (Appendix 1). These probes involved in the concerted process include L187 and L192 in the active site loop, as well as L198, which lies in the dimer interface on one end of the loop (*Figure 1b*). In addition, L101, L121, and L131, which lie in the 'extended loop', are also involved in this process. Of the remaining four probes, three – L74, V79, and L221 – are around the binding site of the cosubstrate mTHF. The active site and extended loops show significant differences at the backbone level between the active and inactive structures (*Figure 1c*), with most of the extended loop lacking electron density in the original inactive structure. In the active conformation, the active site loop is oriented towards the substrate binding site, with the catalytically essential C195 making direct contact with dUMP. In the inactive conformation, on the other hand, the active site loop rearranges via a 180° flip (*Figure 1c*, right) and C195 orients towards the dimer interface. As a result, the interaction between C195 and dUMP that plays a significant role in substrate binding (*Gibson et al., 2008*; *Sapienza and Lee, 2016*) is lost. In addition to the changes in the active site and extended loops, there are small displacements of the backbone throughout the molecule (*Figure 1c*). The fact that these two loops are involved in this concerted process is consistent with this process being exchange between the active and inactive conformations. We have attempted to confirm this by obtaining the chemical shifts of the active and inactive conformations using mutants which stabilize these conformations, but unfortunately these mutants (R163K, M190K, A191K *Gibson et al., 2008*; *Gibson et al., 2008*; *Luo et al., 2011*) either did not express solubly or yielded very poor NMR spectra.

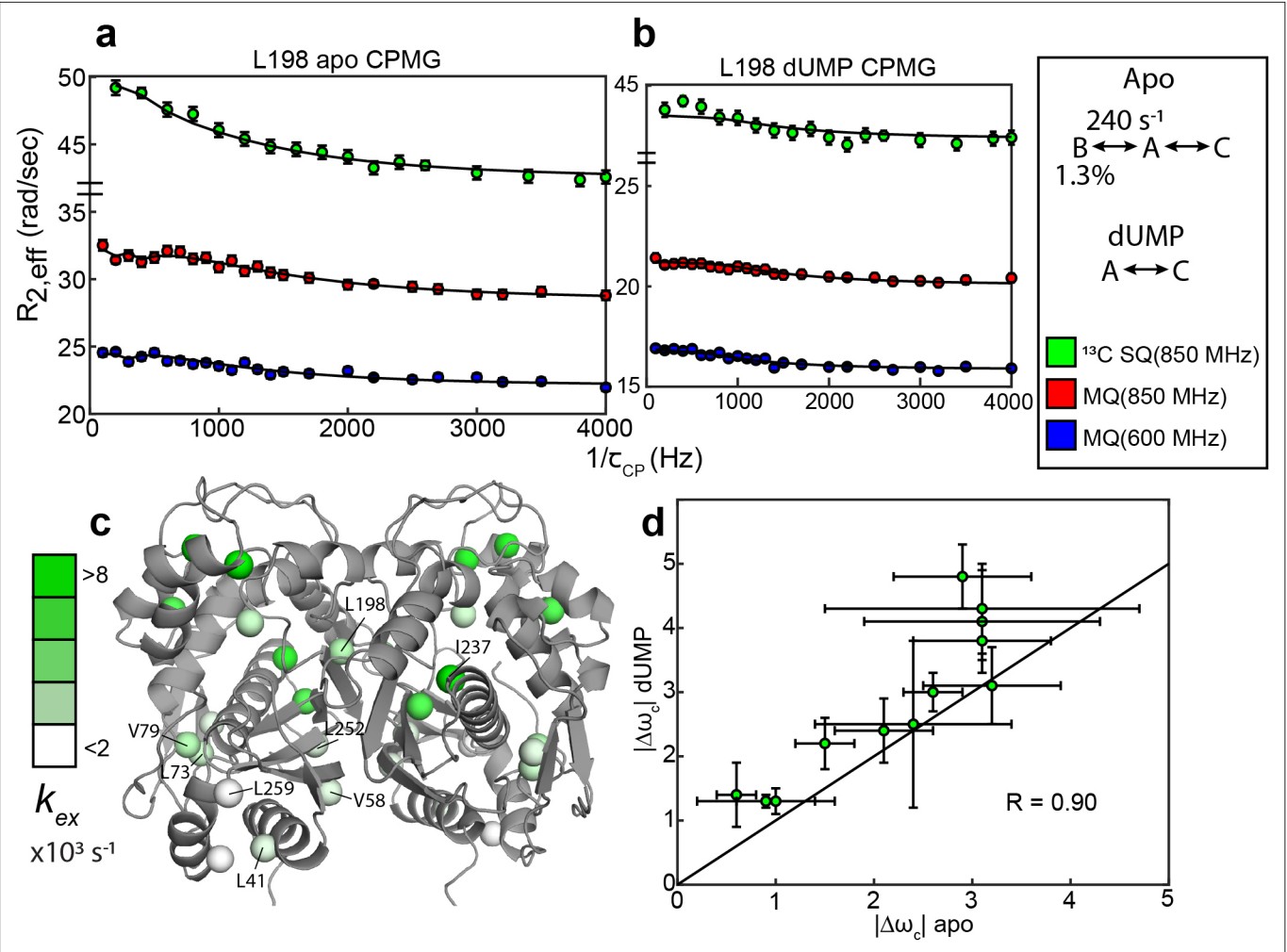

**Figure 2.** Apo and dUMP-bound hTS undergo faster motion with heterogeneous rates and populations. (**a**) CPMG profiles in the global fit, including $^{13}$C SQ at 850 MHz (green), MQ at 850 MHz (red), and MQ at 600 MHz (blue), for L198 in apo hTS using a bifurcated 3-state model (***Figure 2—source data 1–4***). This fit also includes CEST datasets at two spin lock powers (***Figure 1—source data 1 and 2***). Error bars are based on duplicate point RMSD (see Materials & Methods). (**b**) Global fit of three CPMG datasets for L198 in dUMP-bound hTS to a 2-state model (***Figure 2—source data 5–8***). The color scheme is the same as in (**a**). (**c**) Probes with reasonably well-defined exchange rates are displayed on the apo active structure of hTS (PDB ID 5X5A) with the color gradient indicating the $k_{ex}$ value (for 3-state probes, the exchange rate of the faster process is used). (**d**) $^{13}$C $|\Delta\omega|$ values for apo (faster process) and dUMP-bound states are shown. A strong correlation between these values is observed, suggesting that the excited state being accessed is similar in both cases. Error bars are based on fits of 200 Monte Carlo simulated datasets (see Materials & Methods).

The online version of this article includes the following source data for figure 2:

**Source data 1.** Apo hTS LV $^{13}$C MQ CPMG 850 MHz.

**Source data 2.** Apo hTS LV $^{13}$C MQ CPMG 600 MHz.

**Source data 3.** Apo hTS LV $^{13}$C SQ CPMG 850 MHz.

**Source data 4.** Apo hTS I $^{13}$C SQ CPMG 850 MHz.

**Source data 5.** hTS-dUMP LV $^{13}$C MQ CPMG 850 MHz.

**Source data 6.** hTS-dUMP LV $^{13}$C MQ CPMG 600 MHz.

**Source data 7.** hTS-dUMP LV $^{13}$C SQ CPMG 850 MHz.

**Source data 8.** hTS-dUMP I $^{13}$C SQ CPMG 850 MHz.

In addition to the probes involved in the concerted process, there are also 11 probes showing 2-state behavior that have variable exchange rates and populations in apo hTS (***Figure 2***; ***Supplementary file 2***). There appears to be some consistency in exchange rates and populations between nearby probes for these faster motions; for example, L73 and V79 have well-defined exchange rates

and minor state populations that are in excellent agreement, as do L41, L252, and L259 (*Figure 2c*, *Supplementary file 2*). In cases where the minor state population for the faster motion is well-defined, it is almost always very small (0.11 - 0.24%). The three probes which clearly have a larger minor state population are V58 (1.1 ± 0.2%), I237 (4.3 ± 0.9%), and V285 (5% quantile of Monte Carlo simulations is 2%). Interestingly, these localized faster motions generally have $^{13}$C $\Delta\omega$'s that are roughly 3 times as large as those of the concerted, slower motion (*Supplementary file 2*), suggesting considerable change in the local environment of these probes. While the uncertainty in these $\Delta\omega$'s is quite large, even at the lower end of their error bar most of the probes would still have a $\Delta\omega$ at least as large as the largest seen in the slower, concerted process (1.4 ppm).

Finally, there are also two probes showing 3-state exchange that do not appear to be involved in the exchange with the inactive conformation (V58 and L259, *Supplementary file 2*). The parameters for the slower of the two processes for these probes also appear to be distinct from each other.

Global analysis of the CPMG datasets on dUMP-bound hTS (*Figure 2—source data 5–8*) reveal that the slower, concerted process is lost upon substrate binding, but faster motion is still present. Further, comparison of the $^{13}$C $\Delta\omega$'s between apo (fast motion) and dUMP-bound states show a significant correlation despite their uncertainty, suggesting that similar configurations are being accessed (*Figure 2d*). Exchange rates for the faster motion in apo and dUMP-bound states are generally similar as well (*Supplementary files 2 and 3*). It is difficult to comment on whether these faster motions are suppressed or amplified upon dUMP binding due to the relatively large error in the populations in the dUMP-bound analysis, however the smaller number of analyzable probes in the case of dUMP-bound hTS suggests that these motions are at least somewhat suppressed upon substrate binding (*Supplementary file 3*). The loss of the slower, concerted process in dUMP-bound hTS is consistent with this process being exchange between the active and inactive conformations, as the inactive conformation binds dUMP with much lower affinity than the active conformation (*Chen et al., 2017*).

To gain further insights into these faster motions, we performed solvent paramagnetic relaxation enhancement (sPRE) experiments, measuring intensities of the methyl signals in 0 mM and 2 mM gadodiamide (*Figure 3—source data 1*). Gadodiamide is a paramagnetic cosolute, which increases the relaxation rates (decreases signal intensities) of the methyl signals in a manner dependent on the depth, or the distance to the surface, of the methyl probe in the structure. Interestingly, we find that many of the probes around the active site and in the extended loop showing the faster motion have enhanced sPRE (1 – intensity ratio) and thus are more exposed to the solvent than would be expected given their depth in the ground state structure (*Figure 3*). Notably, these data were collected on a truncation of hTS lacking the first 25 residues, termed 'Δ25', which lacks the slower concerted process but retains the faster localized motions (*Appendix 1—figure 2*). This enables us to attribute the observed enhancement in solvent exposure to these localized, faster motions as opposed to the active-inactive equilibrium. In addition, we find that probes possessing this unexpectedly large solvent exposure that did not exhibit exchange in our CPMG and CEST global analysis are nearby probes exhibiting the faster motion. We propose that the non-concerted, faster motion involves localized excursions to 'exposed states', particularly in regions of the protein surrounding the nucleotide and folate binding sites as well as the extended loop. It should be noted that this enhanced exposure of the active site has been seen in other enzymes (*Bernini et al., 2009*). In addition to 1D depth, we have also used 3D atom depth index (*Varrazzo et al., 2005*) for the analysis, yielding very similar results (*Figure 3—figure supplement 1*). Exceptions to this are L67, L198, L279, and V285, which possess the faster motion but do not show enhanced solvent exposure. These probes are marked with asterisks in *Figure 3*. L279 and V285 lie on the opposite end of each protomer relative to the substrate binding site, while L67 and L198 resides at the dimer interface. We note that the lack of enhanced solvent exposure at probes on the dimer interface (*Figure 3b*), the lack of concertedness of the faster motion (*Figure 2c*), and the presence of the faster motion in the dUMP bound enzyme all indicate that this faster motion is not exchange between the dimer and monomeric forms of hTS. The observed enhancement in solvent exposure is consistent with the characteristically large $^{13}$C $\Delta\omega$'s of the faster motion, as this would require relatively large structural changes.

## The active conformation is the ground state of apo hTS

The CPMG and CEST data show that apo hTS undergoes exchange between the active and inactive conformations, and that this exchange process is coupled to dUMP binding. Yet the question

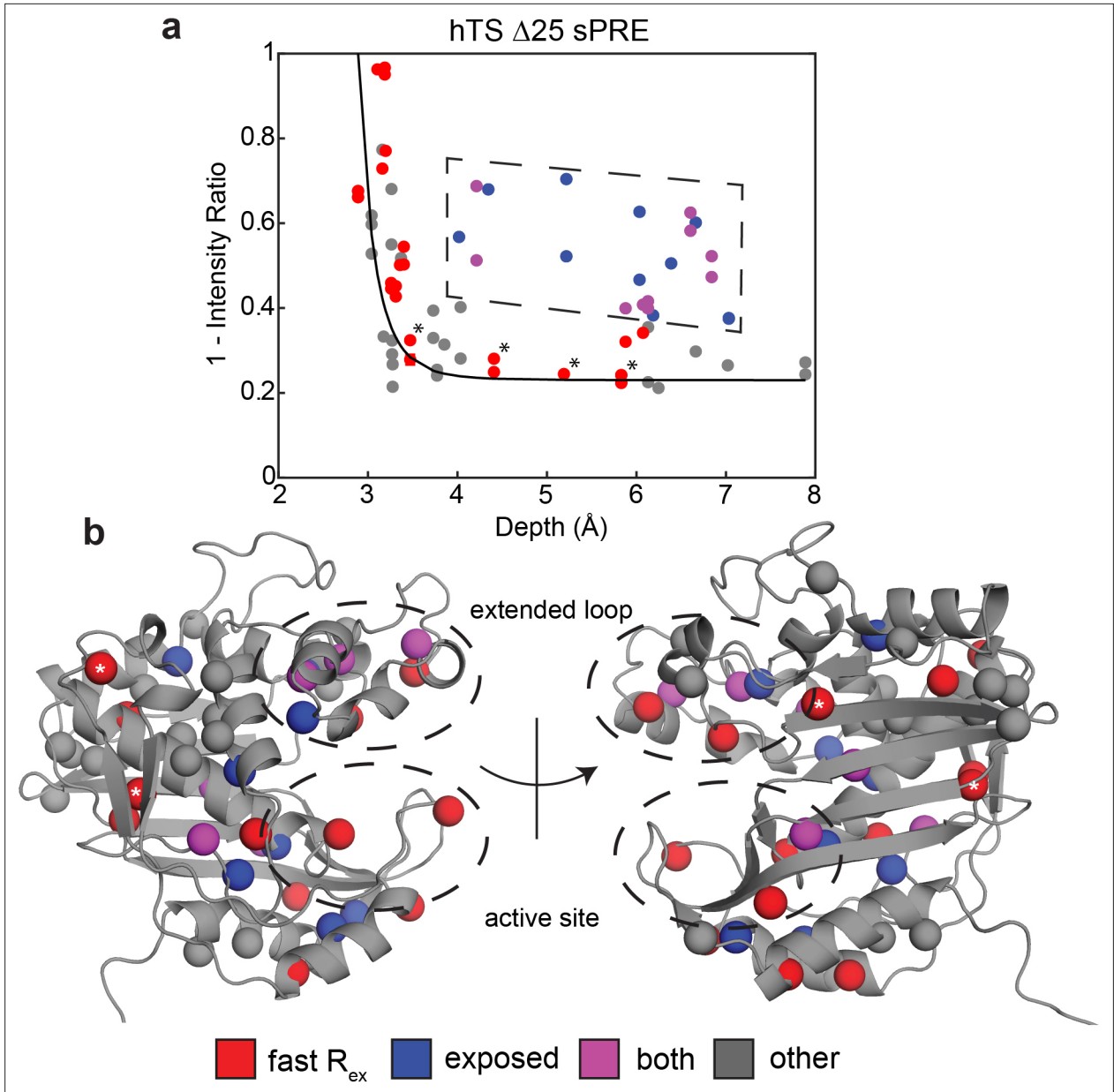

**Figure 3.** Solvent PREs show that many buried faster-motion probes have enhanced solvent exposure. (**a**) Plot of 1 – [Intensity (2 mM Gd)/Intensity (0 mM Gd)] (*Figure 3—source data 1*) as a function of the depth of the probe in the active structure as measured in Δ25 hTS (which possesses faster motion only). Much of the data, to a reasonable approximation, follow a $r^{-6}$ dependence (black curve) analogous to what is seen when the relaxation rates themselves are measured (*Clore and Iwahara, 2009*). However, there are also a number of probes that appear to have a 1 – Intensity Ratio value that is significantly elevated relative to their depth (dashed box, blue). Interestingly, many probes also possess the faster μs-ms motion (purple). Of the rest of the probes possessing the faster motion (red), only four have low 1-Intensity Ratio values (*). (**b**) All the probes from (**a**) are displayed on the active structure, following the same color scheme. Notably, the blue probes (enhanced solvent exposure, but no dispersion seen in our CPMG) tend to be near red/purple probes, suggesting that they may also be involved in the faster processes, or that those processes simply enable solvent to get closer to these probes. In addition, the gray probes (neither fast motion nor enhanced exposure) cluster together. Together, these results suggest that the faster motion generally involves accessing configurations in which these probes are closer to the surface of hTS, which we call 'exposed states'.

The online version of this article includes the following source data and figure supplement(s) for figure 3:

**Source data 1.** Apo Δ25 hTS ILV sPRE.

**Figure supplement 1.** sPRE analysis using SADIC 3D atom depth index yields similar result as 1D depth.

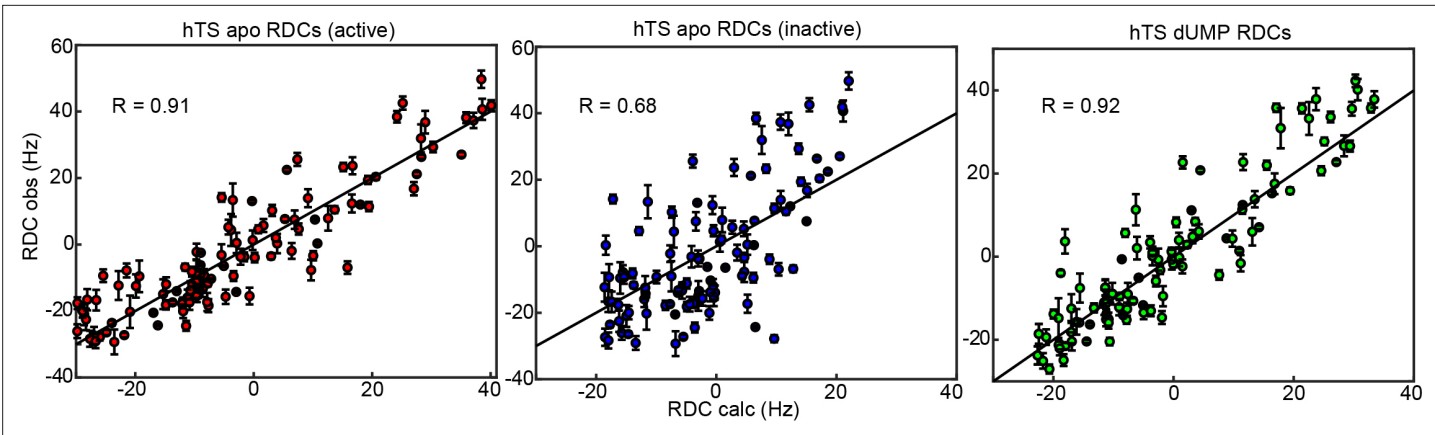

**Figure 4.** RDCs demonstrate that the ground state of apo hTS is the active conformation. Plots show agreement between experimentally observed RDCs (*Figure 4—source data 1–3*) and RDCs calculated from a given structure using a fit alignment tensor. From left to right: Observed apo RDCs with RDCs calculated with the apo active structure (PDB ID 5X5A), observed apo RDCs with RDCs calculated with the apo inactive structure (PDB ID 1YPV), and observed dUMP-bound RDCs with RDCs calculated with the dUMP-bound structure (PDB ID 5X5D). Black lines showing $y = x$ are included to guide the eye. The agreement between the experimental data and the active structure is significantly better than the agreement between the data and the inactive structure; this is supported by the large difference in correlation coefficient between the two cases. In addition, the agreement between the apo RDCs and the active structure is comparable to the agreement between the dUMP-bound RDCs and the dUMP-bound structure. Error bars are based on propagation of 1.5x base plane noise in signal intensities through calculation of the coupling.

The online version of this article includes the following source data and figure supplement(s) for figure 4:

**Source data 1.** Apo hTS $^{15}$N RDC isotropic.

**Source data 2.** Apo hTS $^{15}$N RDC aligned.

**Source data 3.** hTS-dUMP $^{15}$N RDC aligned.

**Figure supplement 1.** $^{13}$C $\Delta\omega$'s for the concerted process do not agree with $^{13}$C chemical shift perturbations upon dUMP binding.

remains as to which of these conformations is the ground state (~98.5%) and which is the excited state (~1.3%). Only if the inactive conformation is the ground state can a conformational selection model contribute significantly to the binding cooperativity, assuming that the inactive conformation binds dUMP much more weakly than the active conformation.

In order to determine which conformation is dominant in solution in apo hTS, we measured residual dipolar couplings (RDCs) on the backbone amides (*Figure 4*, *Figure 4—source data 1 and 2*). These RDCs contain information regarding the orientation of the amide bond vectors; thus, by comparing the experimentally observed RDCs with those calculated from the active and inactive structures using a fit alignment tensor (see Materials and Methods), it can be determined which of the two conformations, if either, is consistent with the data. As a comparison, we also measured RDCs on the backbone amides of dUMP-bound hTS (*Figure 4—source data 3*), where the ground state structure is known. The comparisons show that the active conformation is in significantly superior agreement with the apo hTS data than the inactive conformation (*Figure 4*). Further, we find that the agreement between the apo RDCs and the active structure is just as good as the agreement between dUMP- bound RDCs and the hTS-dUMP structure (also active), having essentially identical correlation coefficients (0.91 and 0.92, respectively). Together, these findings demonstrate that the active conformation is the ground state of apo hTS, indicating that the minor state of the concerted process observed in the CPMG and CEST data is the inactive conformation (1.3%). Consistent with this notion, we find that changes in $^{13}$C chemical shift upon dUMP binding correlate very poorly with, and are generally much smaller in magnitude than, the $^{13}$C $\Delta\omega$'s obtained for the concerted process (*Figure 4—figure supplement 1*). The small population of the inactive conformation in the apo enzyme means that conformational selection contributes negligibly to the substrate binding cooperativity. Since the apo active and dUMP-bound structures are so similar (all atom RMSD = 0.84 Å between 5X5 A and 5X5 D structures), this indicates that there is no significant structural change to the enzyme upon dUMP binding. Consequently, the origin of the dUMP binding cooperativity is likely related to the dynamics of the enzyme.

## Rigidification on the ps-ns timescale of hTS upon dUMP binding correlates with binding cooperativity

Given that we have excluded conformational selection as the source of dUMP binding cooperativity, ps-ns timescale dynamics were considered since these motions are known to impact binding energetics (*Caro et al., 2017*; *Kasinath et al., 2013*). The dynamics of hTS on this much faster timescale were investigated by measuring $^2$H transverse relaxation rates of CHD$_2$ methyl isotopomers in isoleucine, leucine, and valine side chains (*Tugarinov et al., 2005*). From these relaxation rates, we calculate the methyl symmetry axis order parameter, $S^2_{axis}$, which takes on a value between 0 and 1. A value of 0 indicates maximal flexibility of the methyl group, while a value of 1 indicates maximal rigidity. These order parameters are related to the conformational entropy of the molecule; thus, changes in the order parameter upon substrate binding are related to the conformational entropy component of the binding entropy. Reasonable estimates of the change in conformational entropy can be made from these order parameter changes using the 'entropy meter' (*Caro et al., 2017*). This approach assumes that the changes in ps-ns timescale motion seen in the isoleucine, leucine, and valine side chains are representative of all the residues in the protein, and that the overall change in conformational entropy is dominated by the side chains.

We first measured the $^2$H transverse relaxation rates for apo and dUMP bound hTS (*Figure 5—source data 1–2*). The average change in order parameter (dUMP-apo) is 0.027 ± 0.006 (n = 47) (*Figure 5*, *Supplementary files 7 and 8*). Using the entropy meter (see Materials and methods), this corresponds to a $-T\Delta S_{conf}$ of 10 ± 3 kcal/mol. The magnitude of this entropic change is about twice as large as the 4.9 ± 0.3 kcal/mol difference between the entropies of the two dUMP binding events (*Bonin et al., 2019*). This means that if most of the rigidification occurs upon the first binding event, this change in the ps-ns timescale dynamics could be the origin of the positive binding cooperativity.

Probes showing the most rigidification (largest increase in order parameter) are L85, L88, L89, L121, V158, L192, L198, and L221 (*Figure 5c*). Interestingly, much of this rigidification occurs around the binding site of the cosubstrate mTHF (L85, L88, L89, L221). Notably, our ps-ns dynamics analysis lacks probes in the dUMP binding site (*Figure 5b*). In addition, there appears to be considerable colocalization of probes which are rigidified on the µs-ms timescale (i.e. probes which lose the exchange with the inactive conformation) and probes which are rigidified on the ps-ns timescale (*Figure 6*). Of the 6 probes involved in the active-inactive switching (10 total) that are present in the ps-ns dynamics analysis, 4 (L121, L192, L198, L221) show large changes in order parameter ($\left|\Delta S^2_{axis}\right| > 0.06$). The remaining 2 probes (L187, I237), while not displaying a large change themselves, are each in close proximity to a residue possessing a large change in order parameter (V158 and L233, respectively, magenta ovals in *Figure 6*). In addition, there are three probes which have a large change in order parameter that are not in the immediate vicinity of residues involved in the slow motion (L85, L88, and L89, magenta asterisks in *Figure 6*). However, $^1$H 2-plane CPMG data clearly show that V84, and to a lesser extent L85, is dynamic on the µs-ms timescale in apo hTS (*Figure 6—figure supplement 1*). Further, the helix containing these residues (residues 80–93, referred to hereafter as the folate binding helix) connects probes involved in the concerted process, specifically L101 and V79. These observations suggest that L85, L88, and L89 may be involved in the slow motion as well. Finally, 8 of the 9 probes with a large change in order parameter (red/purple in *Figure 6*) showed an increase in order parameter upon dUMP binding (more rigid), the exception being L233.

To further establish the connection between the ps-ns timescale dynamics and the binding cooperativity, we have also investigated these dynamics in two very similar scenarios which lack binding cooperativity. The first is binding of the product TMP to hTS. TMP is very similar to the substrate dUMP, differing only by the addition of a methyl group on the nitrogenous base, yet we have found that hTS binds TMP non-cooperatively (Appendix 1). Further, chemical shift perturbations (CSPs) between the TMP and dUMP-bound forms are generally of a similar (small) magnitude as the CSPs between the dUMP-bound and apo forms (*Figure 7a and b*), suggesting that there is also little structural change between hTS-TMP and hTS-dUMP (there is currently no crystal structure of hTS-TMP). Interestingly, many of the probes having the largest CSPs are clustered at the dimer interface, which is consistent with there being a difference in the intersubunit communication between the TMP and dUMP-bound forms. These probes include L198 and L212 on the beta sheet, as well as L187 from the active site loop and V158 from another loop.

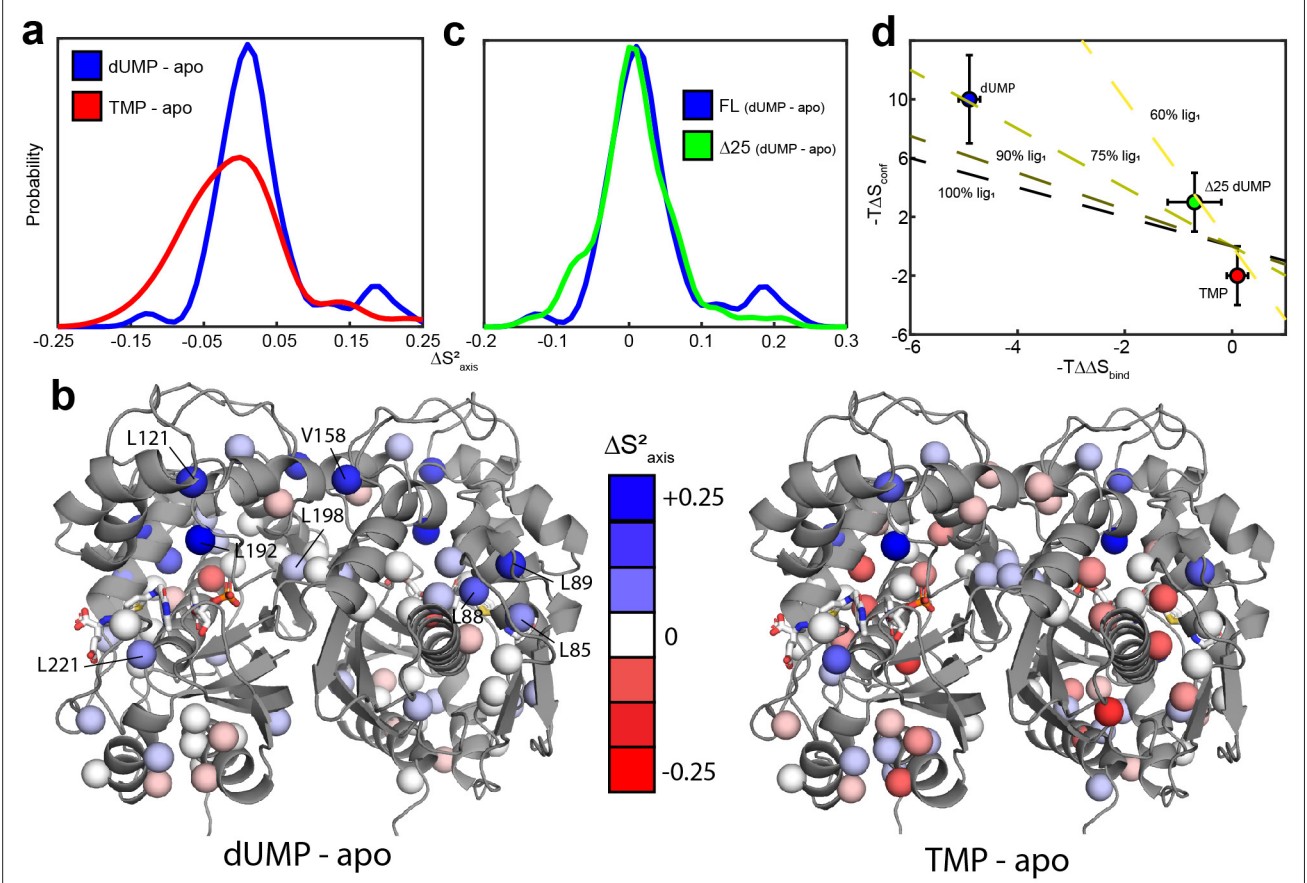

**Figure 5.** hTS is significantly rigidified on the ps-ns timescale upon binding both molecules of dUMP, but not upon binding product TMP nor Δ25 binding dUMP. (**a**) Overlay of kernel distribution fits to $\Delta S^2_{axis}$ values for dUMP-bound – apo hTS (blue, **Figure 5—source data 1–2**) and TMP-bound – apo hTS (red, **Figure 5—source data 1 and 3**). A pronounced shift in the distribution toward higher $\Delta S^2_{axis}$ values (i.e. more rigid) is observed in the case of dUMP binding to hTS. (**b**) Residues included in the analysis are shown on the active structure of hTS (PDB ID 5X5 A), with the $\Delta S^2_{axis}$ values indicated by red-white-blue gradient. Substrates dUMP and mTHF have been added to the structure in sticks. Overall, more blue (increase in $S^2_{axis}$, more rigid) is seen in the case of dUMP binding and more red (decrease in $S^2_{axis}$, more flexible) in the case of TMP binding, consistent with the overlay of distributions in (**a**). (**c**) Overlay of kernel distribution fits to $\Delta S^2_{axis}$ values for dUMP-bound – apo full length hTS (blue) and dUMP-bound – apo Δ25 hTS (green, **Figure 5—source data 4–5**). Again, the distribution is shifted towards higher $\Delta S^2_{axis}$ values in the case of dUMP binding to full length hTS, though the difference is less pronounced in this case. (**d**) There is a correlation between the difference in binding entropies, $-T\left(\Delta S_{bind,2} - \Delta S_{bind,1}\right)$, and the change in conformational entropy upon binding both nucleotide molecules. The X% lig$_1$ lines indicate what the $-T\Delta S_{conf}$ would have to be to produce a given $-T\Delta\Delta S_{bind}$ if all of the change in binding entropy stems from changes in conformational entropy and X% of the $-T\Delta S_{conf}$ occurs upon the first binding event. Error bars are based on fits of Monte Carlo simulated datasets (see Materials & Methods).

The online version of this article includes the following source data for figure 5:

**Source data 1.** apo hTS ILV [2]H transverse relaxation.

**Source data 2.** hTS-dUMP ILV [2]H transverse relaxation.

**Source data 3.** hTS-TMP ILV [2]H transverse relaxation.

**Source data 4.** apo Δ25 hTS ILV [2]H transverse relaxation.

**Source data 5.** Δ25 hTS-dUMP ILV [2]H transverse relaxation.

The small CSPs between hTS-TMP and hTS-dUMP again point to dynamics as being the driving force behind the dUMP binding cooperativity. We have measured [2]H relaxation rates for hTS-TMP as well (**Figure 5—source data 3**), and we find that the average change in order parameter between hTS-TMP and apo hTS is -0.006 ± 0.006 (n = 46) (**Figure 5**, **Supplementary files 7 and 9**). This corresponds to a $-T\Delta S_{conf}$ of -2 ± 2 kcal/mol, meaning that in this case there is actually a slight increase in flexibility of the molecule, as opposed to the rigidification observed with dUMP binding.

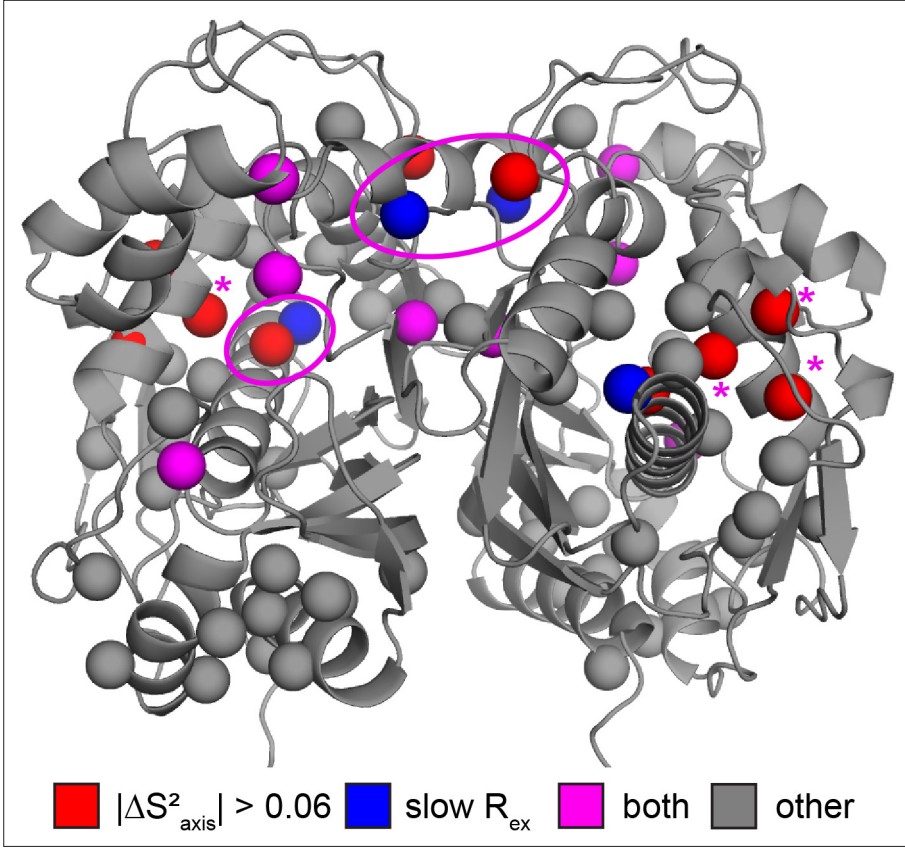

**Figure 6.** Colocalization of probes showing suppression of slow process and large order parameter change upon dUMP binding. Residues having large changes in order parameter are shown in red and residues involved in the slow process on the μs-ms timescale are shown in blue. Probes which meet both of these criteria are shown in purple, while the rest are shown in gray. Of the six probes shown here that are involved in the slow process, four also have large order parameter changes. The two that do not are nearby probes which do have large order parameter changes (purple ovals). While the three remaining red probes, L85, L88, and L89 (purple *), are not near probes identified as being involved in the slow process in our dispersion analysis, the nearby residue V84 clearly shows dynamics on the μs-ms timescale in a [1]H CPMG 2-plane experiment (*Figure 6—figure supplement 1*).

The online version of this article includes the following figure supplement(s) for figure 6:

**Figure supplement 1.** apo hTS [1] H CPMG 2-plane shows μs-ms motion in folate binding helix.

**Figure supplement 2.** ps-ns motions may be necessary, but are not sufficient, to enable concerted μs-ms motion.

The second scenario we consider is dUMP binding to Δ25 hTS. As in the case of TMP binding, this truncated form of hTS binds dUMP non-cooperatively (*Appendix 1—figure 3*), a fascinating result given that the N-terminus is absent in crystal structures of both apo and dUMP bound hTS. It is notable that the non-cooperative *E. coli* enzyme lacks this N-terminal extension, identifying it as a region of interest as a possible determinant of the cooperativity. Δ25 hTS is perhaps an even more striking example than hTS-TMP, as there are essentially no perturbations of the chemical shifts at all to the apo enzyme upon removal of the N-terminus (*Figure 7c*), suggesting that the structures are nearly identical. This suggests once again that differences in the dynamics of these molecules are behind the differences in binding cooperativity.

Consequently, we have measured [2]H transverse relaxation rates for apo and dUMP-bound Δ25 hTS as well (*Figure 5—source data 4–5*). We find that for Δ25 hTS binding dUMP, the average change in order parameter is 0.009 ± 0.004 (n = 76) (*Figure 5*, *Supplementary files 10 and 11*), corresponding to a $-T\Delta S_{conf}$ of 3 ± 2 kcal/mol. While there is still a slight rigidification in this case, the magnitude of the entropic change is less than 1/3 that of dUMP binding to full length hTS. It is worth noting that our analysis of dUMP binding to full length hTS includes only a single residue from the N-terminus itself, V3, as the two N-terminal leucines (L8 and L13) are unassigned. This is to say that the observed

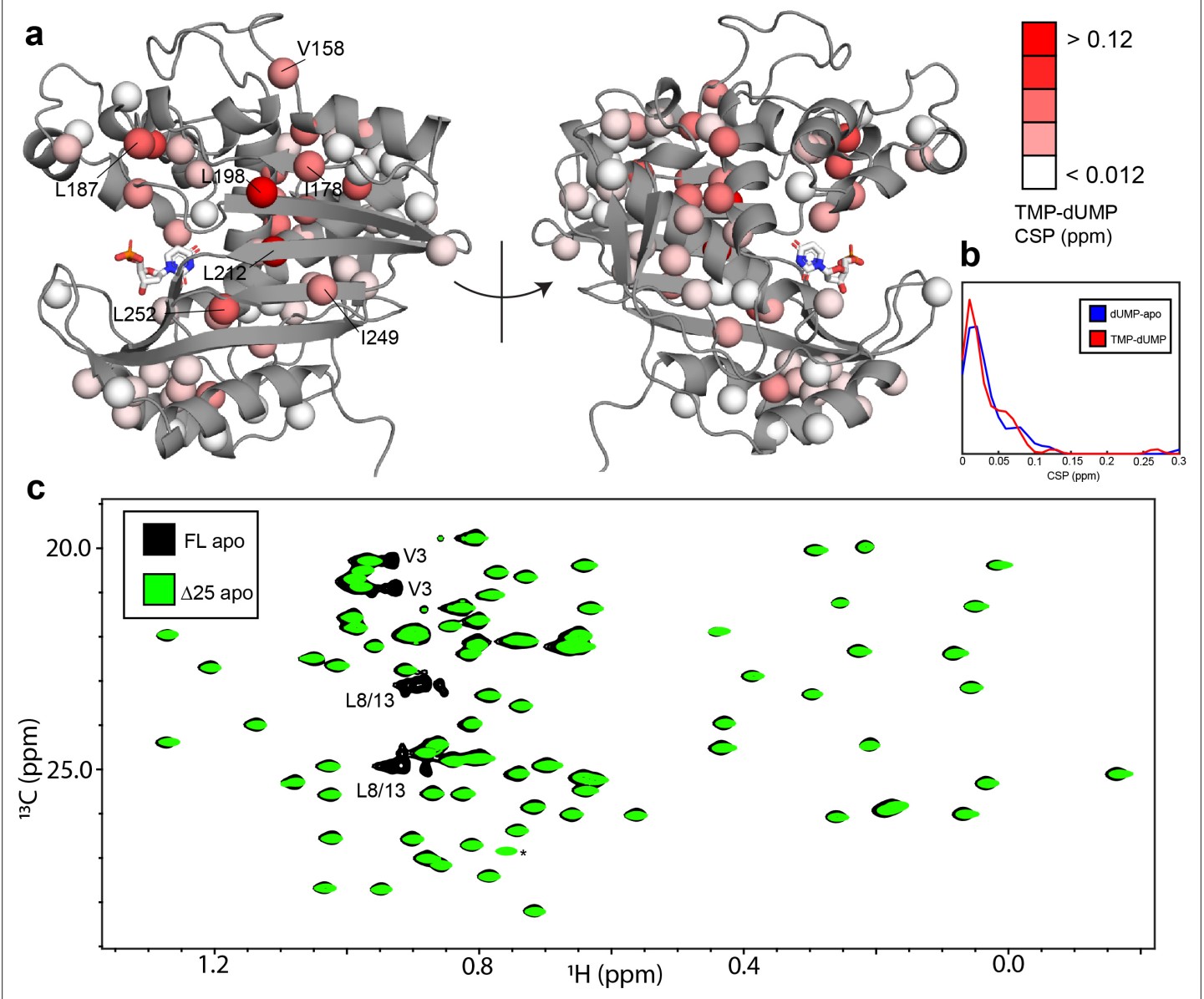

**Figure 7.** hTS-TMP and Δ25 hTS are structurally similar to hTS-dUMP and full length hTS, respectively. CSPs between hTS-TMP and hTS-dUMP,
$\sqrt{\left(\omega_{H,TMP} - \omega_{H,dUMP}\right)^2 + 0.25 * \left(\omega_{C,TMP} - \omega_{C,dUMP}\right)^2}$ , are shown on the apo active structure (5X5A) with the substrate dUMP added in sticks
(**a**).The largest CSPs are often seen at the dimer interface, consistent with the idea that there is a difference in the intersubunit communication in the binding of these two nucleotides. Notably, almost all of the ILV probes on the beta sheet at the interface are in this group, including I178, L198, L212, I249, and L252. Overall, and strikingly, the magnitudes of the CSPs are very similar to that of dUMP binding to the apo enzyme, as can be seen in the distributions of these values (**b**). Since there is little structural change in hTS upon dUMP binding, this suggests that there is also little structural difference overall between hTS-TMP and hTS-dUMP. An overlay of HMQC spectra can be found in *Figure 7—figure supplement 1*. Even more strikingly, there are essentially no perturbations of the chemical shifts between apo full length and Δ25 hTS, as seen in an overlay of HMQC spectra of the Leu and Val methyls (**c**), suggesting that these two structures are virtually identical. The only major difference between these two spectra is the loss of resonances from N-terminal probes V3, L8, and L13 in the Δ25 spectrum, as expected. The signal marked with an asterisk is aliased in the Δ25 spectrum.

The online version of this article includes the following figure supplement(s) for figure 7:

**Figure supplement 1.** Overlay of hTS-dUMP and hTS-TMP ILV HMQC spectra.

differences between full length and Δ25 hTS stem from the N-terminus' impact on the rest of the protein, rather than from the N-terminus itself. These results establish a correlation between the rigidification upon dUMP binding and cooperativity.

## Interactions between N-terminus and protein surface make large contribution to dUMP rigidification

Since the rigidification on the ps-ns timescale upon dUMP binding is likely the source of dUMP binding cooperativity, we sought to determine the origins of this rigidification. An answer is found in considering a thermodynamic cycle consisting of dUMP binding to the full length and Δ25 enzymes (*Figure 8*). We define two paths, A and B, which both begin and end with full length apo and full length dUMP-bound, respectively. Path A contains a single step, which is dUMP binding to the full length enzyme. Path B contains three steps: (1) removal of the N-terminus from the apo enzyme, (2) dUMP binding to the truncated enzyme, and (3) addition of the N-terminus to the truncated dUMP-bound enzyme. The changes in conformational entropy for each of these steps is calculated based on the order parameter values discussed previously. We find that the summed change in conformational entropy across each of these paths agree well with each other, as expected. We also find that the change in conformational entropy upon removal of the N-terminus from the apo enzyme (step 1 of path B) is larger in magnitude than the change in conformational entropy upon addition of the N-terminus to the truncated dUMP bound enzyme (step 3 of path B). This means that the N-terminus has a larger impact on the ps-ns timescale dynamics of the apo enzyme than the dUMP-bound enzyme, suggesting that there are interactions between the N-terminus and the protein surface in apo hTS that are coupled to dUMP binding. These interactions make the apo enzyme more flexible (step 1 of path B has a positive sign). Further, we find that the conformational entropy change associated with this coupling (step 1+step 3 of path B or equivalently path A – step 2 of path B, 6 ± 3 kcal/mol when calculated using probes common to all states) is larger than the conformational entropy change of dUMP binding to the truncated enzyme (step 2 of path B, 3 ± 2 kcal/mol), making this coupling of the N-terminal interactions responsible for the majority of the conformational entropy change on path B. While the uncertainties in the conformational entropy change at each step are fairly large, the change in conformational entropy along path A is clearly larger than step 2 of path B, demonstrating that the N-terminus plays a noticeable role in this change. Together, these results identify the N-terminus of hTS as a key player in the rigidification of the enzyme on the ps-ns timescale upon dUMP binding, suggesting that interactions between the N-terminus and the protein surface, which are diminished/altered upon dUMP binding, contribute significantly to the overall change in conformational entropy. Ironically, it is this flexible N-terminus (which lacks electron density in all crystal structures of hTS) which enables greater rigidification upon dUMP binding.

To gain a deeper understanding of how the N-terminus affects the fast timescale dynamics of hTS, we turn our attention to the differences in order parameter changes between [apo Δ25 and apo full length hTS] and [dUMP-bound Δ25 and dUMP-bound full length hTS] (step 1+step 3 of path B). The location of these differences in $\Delta S^2_{axis}$ , or $\Delta\Delta S^2_{axis}$ , are shown on the hTS structure in (*Figure 8b*). We find that the probes with the largest $\Delta\Delta S^2_{axis}$ are L121, V158, and L233. $\Delta\Delta S^2_{axis}$ values for these probes are nearly identical to their $\Delta S^2_{axis}$ values upon dUMP binding to full length hTS, indicating that the entirety of their ps-ns dynamic response to substrate binding is due to the N-terminus (i.e. the change in order parameter of these probes in step 2 of path B is close to 0). Interestingly, the $\Delta\Delta S^2_{axis}$ values for these probes contain sizable contributions from both the apo and dUMP bound terms, highlighting that despite the larger magnitude of entropy change for step 1 compared to step 3 of path B, the N-terminus still has some effect on the dUMP bound state (i.e. the entropy change of step 3 of path B is nonzero). Additionally, we find that L192 and L221, which are significantly rigidified upon dUMP binding to full length hTS, have $\Delta\Delta S^2_{axis} \cong 0$. This suggests that the entirety of the rigidification of these probes, which lie near the substrate binding site, is due to the direct interactions between dUMP and hTS.

## Discussion

We have shown previously that hTS binds two molecules of dUMP with ~9 fold entropically driven positive cooperativity *Bonin et al., 2019*. Here, we investigate the physical mechanism of this

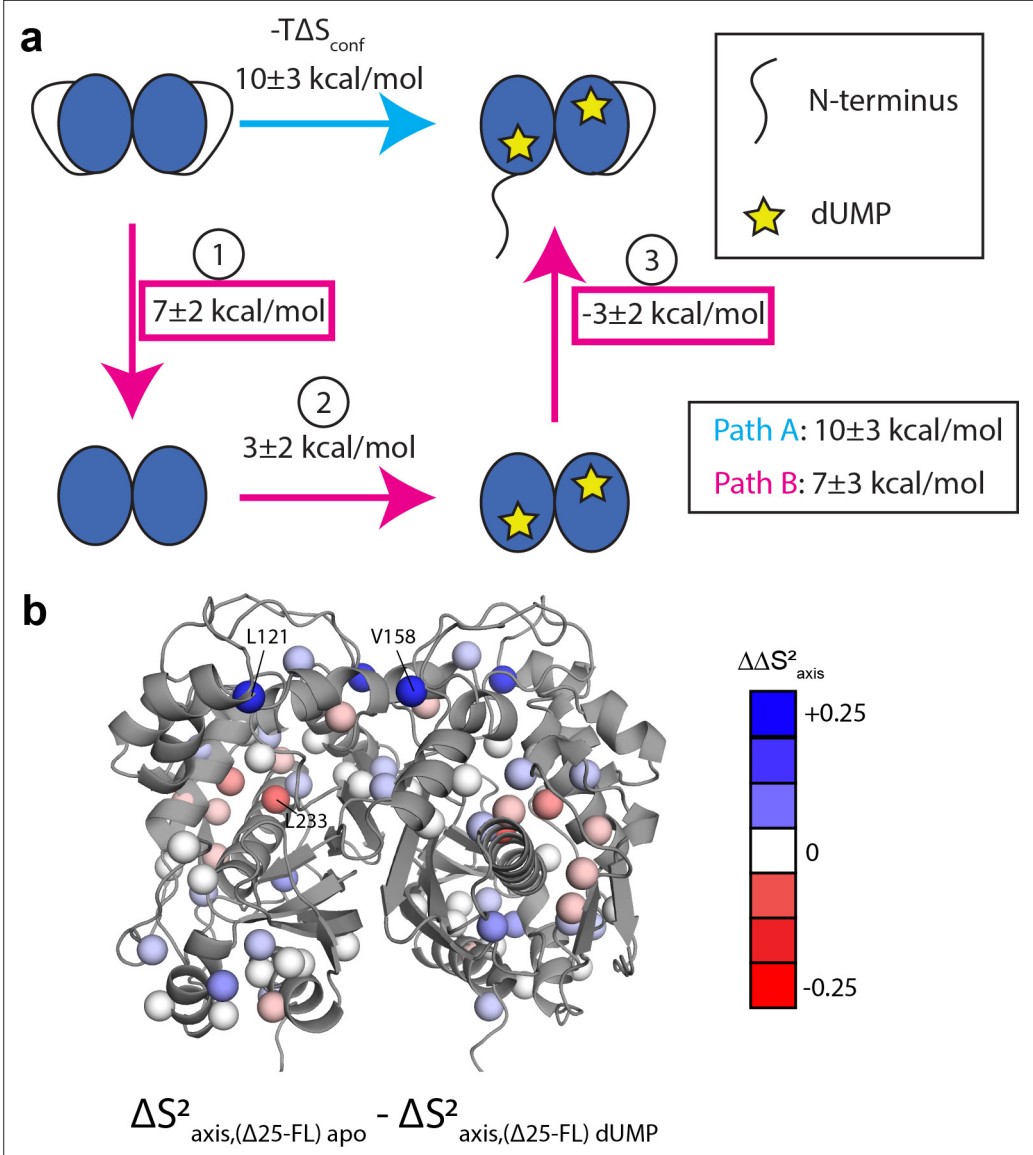

**Figure 8.** Thermodynamic cycle shows that the N-terminus contributes significantly to the rigidification on the ps-ns timescale of hTS upon dUMP binding. (**a**) Thermodynamic cycle involving addition/removal of the N-terminus (full length vs. Δ25 hTS) and dUMP binding. Values are shown next to each step of paths A and B (note that nonidentical sets of probes were used to calculate the changes at each step, leading to the apparent difference between paths A and B). The sum of these values between steps 1 and 3 of path B (or equivalently, the difference between path A and step 2 of path B) represents the coupling between the N-terminus and dUMP binding, and this coupling is larger than the change in step 2 of path B; this identifies the N-terminus as being a key contributor to the overall change in conformational entropy upon dUMP binding. (**b**) Changes in methyl order parameter values between steps 1 and 3 of path B are shown on the active structure, with their values indicated by the red-white-blue gradient. Probes in blue are those which contribute to the rigidification upon dUMP binding that is coupled to the N-terminus (i.e. these probes are more rigidified upon dUMP binding in full length hTS than in Δ25 hTS). Probes showing the greatest rigidification are L121 and V158.

The online version of this article includes the following figure supplement(s) for figure 8:

**Figure supplement 1.** Backbone probes from N-terminus have small perturbations in chemical shift and transverse relaxation rate upon dUMP binding.

cooperativity. The two possible mechanisms are population selection (change in structure) and dynamic allostery (change in dynamics). We also entertain the possibility that there could be contributions from both mechanisms. Given that apo hTS has been crystallized in two distinct conformations, the active and inactive conformations, the population selection mechanism seemed likely. Indeed, this is the mechanism we had previously speculated was behind the binding cooperativity *Bonin et al., 2019*; as discussed in that work, in order for a coupling of the active-inactive equilibrium to dUMP binding to be the source of the cooperativity, the minimum required population of the inactive population in apo hTS needs to be ~89%. Here, we have shown that the ground state of apo hTS is the active conformation, with the inactive conformation being populated to only 1.3%. This is contrary to a previous report, which claimed that the inactive conformation is the ground state at our phosphate condition of 27.5 mM (*Phan et al., 2001*) on the basis of fluorescence measurements. This small population of the inactive conformation in the apo form indicates that the inactive state plays essentially no role in the substrate binding thermodynamics/activity of the enzyme, suggesting that perhaps its only role is to enable binding to the hTS mRNA (*Brunn et al., 2014*). Alternatively, the population of the inactive conformation may be increased by modification of hTS, such as the previously reported phosphorylation of the extended loop (*Fraczyk et al., 2010*).

Consequently, we have probed the dynamics of apo and dUMP-bound hTS on both the μs-ms and ps-ns timescales using NMR spectroscopy in order to gain insight into the mechanism of dUMP binding cooperativity. On the μs-ms timescale, in addition to the exchange with the inactive conformation, we find that apo hTS undergoes faster, localized motion throughout much of the protein. Notably, it is the exquisite sensitivity of the MQ CPMG and CHD$_2$ $^{13}$C CEST experiments that have enabled the discrimination of 2- and 3-state exchange models as well as the identification of the concerted process. Interestingly, there appears to be considerable overlap between regions of the protein undergoing this faster motion and regions of the protein possessing greater solvent exposure than expected given their depth in the hTS active structure. Based on this, we propose that these non-concerted motions involve exchange with 'exposed states'. As discussed previously, we know that the population of the inactive conformation is small enough that the loss of this motion contributes negligibly to the cooperativity. The faster motion likely makes little to no contribution to the cooperativity as well, given that it is present to similar extents in both apo and dUMP bound forms.

On the ps-ns timescale, we find that hTS is significantly rigidified upon binding both molecules of dUMP. This rigidification is especially striking given that our analysis lacks probes within the dUMP binding site. The only ILV side chain within 5 Å of dUMP is V223, which is absent from our analysis of the full length enzyme due to poor signal-to-noise. There are a number of ILV probes which lie just outside this range; these include L192 and L221, which are two of the most rigidified residues in our analysis. This suggests that the observed rigidification may even be underestimating the true magnitude of the effect due to the lack of probes within the dUMP binding site, which are likely also substantially rigidified.

In addition, we observe an apparent colocalization of regions displaying changes in dynamics on the μs-ms and on the ps-ns timescales. This phenomenon has been seen previously (*Henzler-Wildman et al., 2007*; *McDonald et al., 2013*). In McDonald et al., it was reported that changes in ps-ns dynamics were seen largely within the active site and in the allosteric pathway of the protein CheY upon phosphorylation. This is consistent with our results, in which residues showing the largest changes in order parameter are near the dUMP binding site and/or in regions of the protein that are involved in the active-inactive switching. In CheY, McDonald et al. observed that while the active site showed an increase in rigidity, the allosteric pathway showed an increase in flexibility. This contrasts with our results on hTS showing a concomitant increase in rigidity in both areas. This increase in rigidity of regions in which the conformational exchange is suppressed is consistent with the previously proposed idea that faster timescale motion can play a role in enabling slower timescale, concerted motions (*Henzler-Wildman et al., 2007*). Indeed, most of the probes which are a part of the active-inactive switching are quite flexible on the ps-ns timescale in apo hTS, the one exception being L101 (*Figure 6—figure supplement 2*). However, we also find that in hTS Δ25, which lacks the active-inactive exchange, there is little to no change in the order parameter of these residues (*Figure 6—figure supplement 2*). This demonstrates that the presence of ps-ns motion on its own is not sufficient to produce slower timescale, concerted motions.

To assess the plausibility of this rigidification upon dUMP binding being the source of the cooperativity, we first used the 'entropy meter' (*Caro et al., 2017*) to quantify a reasonable estimate of the change in conformational entropy associated with the order parameter changes we observed. Verifying that this is the case would require us to make these same measurements on a sample of hTS that is bound by only a single molecule of dUMP, which is a work in progress. We have recently developed an approach that has enabled making this kind of sample of a different dimeric system (*Sapienza et al., 2021*).

In the absence of measurements on the singly-bound state of hTS, we instead support the idea that the changes in ps-ns motions are the source of the binding cooperativity by comparing these changes to those of hTS binding TMP and Δ25 hTS binding dUMP, both of which are non-cooperative. hTS-TMP and apo Δ25 hTS are very similar structurally to hTS-dUMP and apo full length hTS, respectively, suggesting that the differences in binding thermodynamics are largely due to differences in the dynamics of the enzyme. Indeed, both of these non-cooperative cases lack the pronounced rigidification observed with dUMP binding to full-length hTS. In the case of the product TMP binding to hTS, we observe an entropic free energy change of -2 ± 2 kcal/mol, indicating that there is actually a slight increase in the flexibility of the enzyme. The difference in the response of the ps-ns dynamics to dUMP and TMP binding is especially striking considering that TMP differs from dUMP only in the addition of a methyl group on the nitrogenous base. Clearly, hTS has been designed to be exquisitely sensitive to the presence of the TMP methyl group in terms of its dynamics. In the absence of this methyl group (dUMP), the enzyme is rigidified, particularly in the folate binding helix. This helix contains F80, which makes direct hydrophobic contact with the cosubstrate mTHF. Consequently, we hypothesize that this rigidification may serve as a means to preorganize the mTHF binding site upon dUMP binding. Consistent with this idea, we observe rigidification of L221 as well, which also makes hydrophobic contact with mTHF. In the presence of this methyl group (TMP), there is a widespread, slight increase in the flexibility of the enzyme, especially relative to the dUMP bound form. In particular, the folate binding helix is noticeably more flexible in the TMP bound form than the dUMP bound form. The increase in conformational entropy of this region, and of the enzyme as a whole, in the TMP bound form may aid in the release of DHF (the folate product of the reaction) after the reaction has occurred. Also of note is V158, residing on the dimer interface, which has the largest difference in order parameter between the dUMP and TMP bound states.

In the case of the truncated enzyme, Δ25, binding dUMP, there is an overall rigidification of the enzyme, but the magnitude is significantly smaller than that observed in the full length enzyme. This difference in the response of the ps-ns dynamics of hTS to dUMP binding between full length and Δ25 indicates that the N-terminus is somehow involved in the rigidification. Little is known structurally about the N-terminus, given that it is absent in all crystal structures of hTS, although it has been shown to play a role in the enzymatic activity (*Huang et al., 2010*). Backbone amide probes from the N-terminus show only very subtle perturbations of chemical shift and transverse relaxation rate upon dUMP binding, suggesting that dUMP binding has little impact on the N-terminus itself (*Figure 8—figure supplement 1*). Further, the very low transverse relaxation rates of the N-terminal residues indicate that this region is highly flexible (i.e. disordered) in both apo and dUMP bound states, consistent with the crystallography. In addition, $C^\alpha$ chemical shifts from the N-terminus change negligibly upon dUMP binding (*Figure 8—figure supplement 1*), further indicating that there is no noticeable increase in secondary structure propensity. While a disordered region has not been implicated in the previously described examples of dynamic allostery (*Capdevila et al., 2017*; *Petit et al., 2009*; *Popovych et al., 2006*), an allosteric mechanism utilizing intrinsically disordered regions has been described previously (*Hilser and Thompson, 2007*). However, this mechanism involves the disordered region folding upon binding, which is not the case in hTS. The N-terminus' contribution to the rigidification of hTS upon dUMP binding, presumably in the form of transient interactions between the N-terminus and the protein surface that are perturbed by substrate binding, thus represents a novel mechanism of dynamic allostery involving an unexpected role for a disordered region.

Our finding of the N-terminus' considerable role in the ps-ns dynamics of the protein as a whole raises two major questions: (1) where does the N-terminus interact with the rest of the protein, and (2) why are these interactions coupled to dUMP binding? One possibility is that the N-terminus interacts with the dUMP binding site itself, which would easily explain why these interactions are coupled to the binding. Another possibility is that the N-terminus interacts in the vicinity of L121 and V158, the

probes which contribute most to the coupling entropic free energy. Interestingly, in a crystal structure of the inactive conformation (PDB ID 1YPV), there is electron density ascribed to V3 that is within 10 Å of V158, consistent with this idea. Further, we observe a small change in the order parameter of V3 upon dUMP binding to full length hTS (*Supplementary files 7 and 8*). If this model is indeed correct, the question remains as to why these interactions are coupled to dUMP binding. Given that there is no significant structural change upon dUMP binding, the coupling is likely dynamic in nature. However, V158 (as well as L121) has essentially no change in order parameter upon dUMP binding in Δ25 hTS, indicating that there is no dynamic coupling on the ps-ns timescale between this probe and the active site unless the N-terminus is present. Further work is required to answer these questions definitively.

This work also highlights the prominent role of the 3 insertions in hTS, relative to the *E. coli* enzyme, in the dynamics of the protein. These insertions are the N-terminus, part of the extended loop (residues 115–126), and part of a second loop (residues 146–153). TS is a highly conserved enzyme, with the human and *E. coli* forms having 55% identity in amino acid sequence. Indeed, these insertions are the only major structural difference between the enzymes. Interestingly, the *E. coli* enzyme, like Δ25 hTS, lacks both dUMP binding cooperativity (*Sapienza et al., 2015*) and the inactive conformation. We have found that the N-terminus and the extended loop are involved in the active-inactive switching; L101, L121, and L131 from the extended loop are part of this concerted process, and this motion is absent in the N-terminally truncated enzyme. In addition, the N-terminus plays a significant role in the rigidification of hTS on the ps-ns timescale upon dUMP binding. L121, from the extended loop, as well as V158, the only ILV probe from the second loop, are two of the most rigidified probes. Further, the rigidification of these probes upon dUMP binding only occurs when the N-terminus is present, indicating that the dynamics of these three regions are coupled. To somewhat oversimplify, it could be said that these three flexible insertions introduce dUMP binding cooperativity into TS. Evolution may have utilized such an approach to introduce allosteric properties into other proteins as well.

In summary, we have shown apo hTS exists almost entirely in the active conformation, populating the inactive conformation to only 1.3%. This means that there is essentially no change in the structure of hTS upon dUMP binding, indicating that the mechanism of the entropically-driven binding cooperativity is primarily dynamic in nature. Indeed, we have determined that there is substantial rigidification of hTS on the ps-ns timescale upon dUMP binding, primarily around the binding site and in regions involved in the active-inactive switching. Further, we find that this rigidification is absent in two cases of hTS binding nucleotide non-cooperatively (TMP binding to full length hTS and dUMP binding to Δ25 hTS). Finally, we have shown that this rigidification is largely due to interactions between the intrinsically disordered N-terminus of hTS (a 25 residue segment that is absent in all crystal structures of the enzyme) and the rest of the protein which are coupled to dUMP binding, a novel mechanism of dynamic allostery. Together, these results provide a rare depth of insight into the mechanism of substrate binding cooperativity in hTS.

## Materials and methods
### Sample preparation
Samples of hTS were expressed and purified as described previously (*Bonin and Lee, 2021*), with modifications made for the specific methyl labeling utilized here. Briefly, a pET21a vector including hTS with an N-terminal 6 x histidine tag was transformed into BL21 (DE3) cells lacking the *rne131* gene (Invitrogen, Carlsbad, CA). After an adaptation period (*Bonin and Lee, 2021*), the cells were grown in M9 media made with $D_2O$ (99.8%, Sigma-Aldrich, St. Louis, MO/Cambridge Isotope Laboratories, Inc, Tewksbury, MA), 1 g/L $^{15}NH_4Cl$ (Sigma-Aldrich, St. Louis, MO/Cambridge Isotope Laboratories, Inc, Tewksbury, MA) and 2 g/L U-[$^2H$] glucose (Sigma-Aldrich, St. Louis, MO/Cambridge Isotope Laboratories, Inc, Tewksbury, MA). Cells were grown up to an $OD_{600}$ of 0.65 at 37 °C, at which time one of two Ile and Leu/Val precursors was added: 60 mg/L 2-ketobutyric acid-4-$^{13}C$,3,3-$d_2$ sodium salt hydrate (Ile) and 100 mg/L 2-keto-3-(methyl-$d_3$)-butyric acid-4-$^{13}C$, 3-d sodium salt (Leu/Val) for {I ($^{13}C_\delta H_3$), L($^{13}C_\delta H_3$,$^{12}C_\delta D_3$), V($^{13}C_\gamma H_3$,$^{12}C_\gamma D_3$)} U-[$^2H$,$^{15}N$] hTS (referred to hereafter as ILV $CH_3$ hTS) or 60 mg/L alpha-ketobutyric acid-4-$^{13}C$,3,3,4,4-$d_4$ sodium salt (Ile) and 100 mg/L alpha-ketoisovaleric acid-3-methyl-$^{13}C$,3-methyl-$d_2$,3,4,4,4-$d_4$ sodium salt (Leu/Val) for {I ($^{13}C_\delta HD_2$), L($^{13}C_\delta HD_2$,$^{12}C_\delta D_3$), V($^{13}C_\gamma HD_2$,$^{12}C_\gamma D_3$)} U-[$^2H$,$^{15}N$] hTS (referred to hereafter as ILV $CHD_2$ hTS). One hour after the addition of the precursors, IPTG was added to 1 mM, and protein was expressed at 20 °C for 12 hr. For samples

without the ILV labeling (U-[$^2$H,$^{15}$N] hTS, referred to hereafter as $^{15}$N hTS), IPTG is simply added at an OD$_{600}$ of 1.0, followed by expression at 20 °C.

The purification workflow was as described previously (**Bonin and Lee, 2021**). Notably, cleavage of the 6 x histidine tag leaves an extra glycine and alanine at the N-terminus. Protein was concentrated to ~0.2 mM (dimer) for NMR study, as determined by 280 nm absorbance using $\varepsilon_{280} = 117,040 \frac{L}{mol*cm}$ (**Bonin et al., 2019**). The final conditions for NMR study were 27.5 mM Na$_2$HPO$_4$, 20 mM NaCl, 0.02% NaN$_3$, 5 mM DTT, pH 7.5 (NMR buffer) with D$_2$O added to 5%, and a temperature of 25 °C. For samples in D$_2$O, the NMR buffer was prepared in 99.8% D$_2$O and pD (uncorrected) was 7.1 (**Covington et al., 1968**). The sample was exchanged into the D$_2$O buffer using a NAP-5 Sephadex G-25 column (GE Healthcare, Chicago, IL). For dUMP and TMP (Sigma-Aldrich, St. Louis, MO) bound samples, nucleotide was added to a ratio of ~10:1 nucleotide:dimer as determined by 260 nm absorbance using $\varepsilon_{260} = 9660 \frac{L}{mol*cm}$ (dUMP) or $\varepsilon_{260} = 7936 \frac{L}{mol*cm}$ (TMP).

## Evaluation of apo hTS ground state structure using RDCs

A 0.2 mM sample of apo $^{15}$N hTS in NMR buffer (5% D$_2$O) was prepared as described. Residual dipolar couplings (RDCs) of backbone amides were determined using the ARTSY method (**Fitzkee and Bax, 2010**). First, a measurement of the isotropic coupling (J coupling) was made, after which Pf1 phage (ASLA Biotech, Riga, Latvia) was added to 7 mg/mL, resulting in a 4.7 Hz splitting of the D$_2$O line. A second coupling measurement was made in this aligned condition (J+RDC), and the RDC was calculated by taking the difference between aligned and isotropic couplings. dUMP was then added to the aligned sample to 2.5 mM, and the aligned coupling measurement was repeated. RDCs for dUMP bound hTS were computed using the J couplings from apo hTS. The error in signal intensities was taken to be 1.5 x base plane noise based on the duplicate point RMSD observed in pseudo-3D datasets. These measurements were made on a Bruker Avance III HD 850 MHz spectrometer with a TCI H-C/N-D 5 mm cryoprobe. Additional details regarding acquisition parameters for these and all other NMR experiments can be found in **Supplementary file 5**. Unless otherwise stated, all NMR data were processed using NMRPipe (**Delaglio et al., 1995**) and signal intensities were obtained using NMRViewJ (**Johnson and Blevins, 1994**).

RDC data were analyzed using the DC web server. The alignment tensor was fit (order matrix method) using the observed RDCs and a structure of hTS (5X5A: active, 1YPV: inactive, 5X5D: dUMP bound), with the calculated RDCs being computed from the alignment tensor and the structure. Residues from the flexible N and C termini (1-29, 301-313) were excluded from this analysis, as were residues 108–129, which lack electron density in the inactive structure.

## Characterization of µs-ms motions using CPMG and CEST

For MQ CPMG (**Korzhnev et al., 2004**) experiments, a 0.2 mM sample of LV CH$_3$ apo hTS in NMR buffer (99.8% D$_2$O) was prepared as described. Datasets were collected at 850 and 600 MHz. dUMP was added to 2.2 mM and datasets were recollected. For $^{13}$C SQ CPMG (**Lundström et al., 2007**) experiments, a 0.3 mM sample of ILV CH$_3$ apo hTS in NMR buffer (99.8% D$_2$O) was prepared as described. Datasets were collected at 850 MHz with $^{13}$C carrier frequencies at 24 ppm (Leu/Val) and 14 ppm (Ile). dUMP was added to 2.8 mM and datasets were recollected. For $^{13}$C CEST (**Rennella et al., 2015**) experiments, a 0.3 mM sample of ILV CHD$_2$ apo hTS in NMR buffer (99.8% D$_2$O) was prepared as described. Datasets were collected at 600 MHz using spin lock powers of 41 and 25 Hz as determined by calibration via nutation experiment. 500 Hz $^2$H decoupling was applied during the relaxation period.

For CPMG datasets, $R_{2,eff}$ values were calculated from peak intensities (**Source code 1**) using the following equation:

$$R_{2,eff}\left(plane\ i\right) = -\frac{1}{T_{relax}} \ln\left(\frac{Int\left(plane\ i\right)}{Int\left(ref\ plane\right)}\right)$$

where $T_{relax}$ is the relaxation time and *plane i* is the *i*th plane with 180° pulses applied at frequency $\upsilon_{cpmg}\left(i\right)$. The error in signal intensity was taken to be the duplicate point RMSD for the CPMG datasets and 1.5 x base plane noise for the CEST datasets. The three sets of $R_{2,eff}$ values (MQ 850 MHz, MQ 600 MHz, and $^{13}$C SQ 850 MHz), along with two sets of normalized intensities from the CEST experiments (41 and 25 Hz spin lock) were globally analyzed for each probe (for dUMP bound hTS, the

analysis includes only the CPMG data). Initially, data were fit to a 2-state model (*Source code 8 and 10*). For some probes, however, the data were not well described by a 2-state model, and a bifurcated 3-state model (B↔A↔C, *Source code 2–9*) was employed in which the A↔B process was concerted. The $R_{2,eff}$ and normalized intensity values are calculated from the exchange parameters using the Bloch-McConnell equations (see Appendix 1 for relevant matrices). For a detailed consideration of alternative 3-state models, see Appendix 1. For the 3-state model fits, individual rates were fit as opposed to exchange rates and populations ($k_{ab}$, $k_{ba}$, $k_{ac}$, $k_{ca}$). The reason for this is that when there is 3-state exchange where only one of the two processes is concerted, only the rates (and not the populations) are identical amongst the probes involved in the concerted process. For probes involved in the concerted process, $k_{ab}$, $k_{ba}$, and $\Delta\omega_{C,ab}$ were fixed as values obtained from a global 2-state fit of the CEST data alone (*Source code 11–13*), as 3 (2)-state fits of the CPMG and CEST data without any fixed parameters were found to give the same answer. The one exception to this is I237, whose $\Delta\omega_{C,ab}$ was found to change noticeably in a 3-state fit of the CPMG and CEST data where no parameters were fixed. ${}^1$H $\Delta\omega$'s were fixed at 0 ppm unless a nonzero value was required to obtain a reasonable fit to the data (*Korzhnev et al., 2004*). Geminal methyls of the same residue were fit globally. The errors in fit parameter values are taken as the standard deviation of the values obtained in fits of 200 Monte Carlo simulations of the data.

## Assessing exposure of fast motion probes using solvent PREs

A 0.1 mM sample of LV $CH_3$ apo Δ25 hTS in NMR buffer (99.8% $D_2O$) was prepared as described. HMQC spectra were recorded with 0 and 2 mM gadodiamide (APExBIO, Houston, TX). These spectra were collected on an 850 MHz spectrometer. The intensity ratio is defined as:

$$Int\,Ratio = \frac{Int\,(2\,mM\,Gd)}{Int\,(0\,mM\,Gd)}$$

The error in signal intensity was taken to be 1.5 x base plane noise. Residue depths were calculated from the 5X5A structure using EDTSurf (*Xu et al., 2013*).

## Characterization of ps-ns motions using ${}^2$H relaxation

A 0.2 mM sample of ILV $CHD_2$ apo hTS in NMR buffer (5% $D_2O$) was prepared as described. All ${}^2$H $R_2$ relaxation (*Tugarinov et al., 2005*) data were collected on a 600 MHz spectrometer. In the absence of duplicate points (*Supplementary file 5*), the error in peak intensity was estimated as 1.5 x the base plane noise. dUMP was added to 2 mM to the apo sample, and ${}^2$H relaxation data were collected as for apo. For the product-bound state, a 0.2 mM sample of ILV $CHD_2$ hTS-TMP in NMR buffer (5% $D_2O$, 2 mM TMP) was prepared as described. ${}^2$H relaxation data were collected as for apo and dUMP bound forms. For the truncated enzyme, a 0.2 mM sample of ILV $CHD_2$ apo Δ25 hTS in NMR buffer (99.8% $D_2O$) was prepared as described. After collection of the data on the apo enzyme, dUMP was added to 1.7 mM and the data were recollected.

Peak intensities were normalized relative to the first point ($T_{relax}$ = 0 ms) and fit to a monoexponential decay to obtain the relaxation rate, $R_{2,^2H}$. The errors in transverse relaxation rates were taken as the standard deviation of the values obtained in fits of 200 Monte Carlo simulations of the data (derived from errors estimated from duplicate points or baseplane noise). The ${}^2$H relaxation rate, along with the rotational correlation time, $\tau_c$, of the molecule (see below), was then used to calculate the methyl rotation axis order parameter, $S_{axis}^2$, using the following equation:

$$R_{2,^2H} = \frac{1}{80}\left(\frac{e^2qQ}{\hbar}\right)^2 S_{axis}^2\tau_c$$

where $\frac{e^2qQ}{\hbar}$ is the quadrupolar coupling constant with a value of 167 kHz (*Tugarinov et al., 2005*). Changes in conformational entropy were calculated from changes in the methyl symmetry axis order parameter using the following equation:

$$-T\Delta S_{conf} = -298.15 * s_d \left\langle \Delta S_{axis}^2 \right\rangle N_\chi$$

where $\left\langle \Delta S_{axis}^2 \right\rangle$ is the average change in order parameter, $N_\chi$ is the number of side-chain dihedral angles in the protein (1136 used here, calculated using residues 26–307), and $s_d = -0.0011 \frac{kcal}{mol*K}$ is the slope of the entropy meter (*Caro et al., 2017*). Only probes with errors in $\Delta S_{axis}^2 < 0.07$ were included in the analysis.

## Structural similarity between full length and Δ25 hTS shown by HMQC spectra

Samples of 0.1 mM ILV CH$_3$ apo hTS and 0.2 mM LV CH$_3$ Δ25 apo hTS in NMR buffer (99.8% D$_2$O) were prepared as described. HMQC (*Tugarinov et al., 2003*) spectra were collected on an 850 MHz spectrometer.

## Determination of rotational correlation time

Rotational correlation times for the various states of hTS were determined by $^{15}$N relaxation (*Lakomek et al., 2012*) collected on samples of closely matched concentration. For the apo form, data were collected on separate samples at two fields, 600 MHz [500 MHz]. A 0.2 [0.2] mM sample of $^{15}$N apo hTS in NMR buffer (5% D$_2$O) was prepared as described. $T_1$, $T_{1\rho}$, and heteronuclear NOE datasets were collected at both fields. All $T_{1\rho}$ datasets, unless otherwise stated, were collected with 1.5 kHz spin lock power as determined by nutation experiment. For the dUMP bound form, $T_1$ and $T_{1\rho}$ datasets were collected at two fields, 600 MHz [500 MHz], and a single heteronuclear NOE dataset was collected at 600 MHz. A 0.25 [0.25] mM sample of $^{15}$N hTS-dUMP (2.5 mM dUMP) in NMR buffer (5% D$_2$O) was prepared as described. For the TMP bound form, $T_1$, $T_{1\rho}$, and heteronuclear NOE datasets were collected at 600 MHz. A 0.2 mM sample of $^{15}$N hTS-TMP (2.8 mM TMP) in NMR buffer (5% D$_2$O) was prepared as described. The $T_{1\rho}$ dataset was collected with 2 kHz spin lock power as determined by nutation experiment. For the apo form of Δ25 hTS, $T_1$, $T_{1\rho}$, and heteronuclear NOE datasets were collected at 600 MHz. A 0.2 mM sample of $^{15}$N apo Δ25 hTS in NMR buffer (5% D$_2$O) was prepared as described. For the dUMP bound form of Δ25 hTS, $T_1$ and $T_{1\rho}$ datasets were collected at 600 MHz. A 0.2 mM sample of $^{15}$N Δ25 hTS-dUMP (2 mM dUMP) in NMR buffer (5% D$_2$O) was prepared as described.

Intensities from $T_1$ datasets were normalized relative to the $T_{relax} = 0$ s point and fit to a monoexponential decay to obtain the longitudinal relaxation rate, $R_1$. Intensities from $T_{1\rho}$ datasets were normalized relative to the $T_{relax} = 0$ s point and fit to a monoexponential decay to obtain $R_{1\rho}$. Transverse relaxation rates, $R_2$, were then calculated from $R_{1\rho}$, $R_1$, and tilt angle $\theta$ using the equation:

$$R_2 = \frac{R_{1\rho}}{\sin^2 \theta} - \frac{R_1}{\tan^2 \theta}$$

The heteronuclear NOE was calculated as $\frac{Int(saturated)}{Int(ref)}$. Error in signal intensity was taken to be the mean residual of the best fit for $T_1$ and $T_{1\rho}$ datasets, and 1 x base plane noise for heteronuclear NOE datasets. The ROTDIF web server (*Berlin et al., 2013*) was used to fit $\tau_c$ values for these different states of hTS given the relaxation rates and the 5X5A (apo) or 5X5D (dUMP, TMP) structure. The criteria for inclusion of residues in this fit were as follows: $R_1$ and $R_2$ values were subject to $\log_2$ transformation to normalize their distributions, after which the mean and standard deviation of the normalized distributions were calculated (residues from flexible N- and C-termini were not included in these calculations). Residues with $\log_2 R_1$ and/or $\log_2 R_2$ values outside one standard deviation of the mean were excluded from the analysis. In addition, in cases where NOE data were present, any residue with an NOE <0.65 were excluded from the analysis. In the case of apo Δ25 hTS, residues with NOE <0.6, as well as residues identified by ROTDIF as possessing $R_{ex}$, were excluded from the analysis. Notably, specific criteria for which residues to include, as well as the type of diffusion tensor used (isotropic, axially symmetric, fully anisotropic) had very little impact on the fit $\tau_{c,eff}$. Fits to axially symmetric and fully anisotropic models showed only slight anisotropy to the diffusion tensor, justifying use of the isotropic model for analysis of the $^2$H relaxation data. Note that this rotational correlation time is determined in 5% D$_2$O; for samples in 99.8% D$_2$O, the correlation time is corrected based on the relative viscosities of D$_2$O and H$_2$O:

$$\tau_{c,D_2O} = 1.235 * \left( \frac{\tau_{c,5\% \, D_2O}}{0.95 + 0.05 * 1.235} \right)$$

where $\tau_{c,D_2O}$ and $\tau_{c,5\% D_2O}$ are the rotational correlation times in 99.8% and 5% $D_2O$, respectively, and 1.235 is the ratio of the viscosities of $D_2O$ and $H_2O$ at 25 °C (*Baker, 1936*). Rotational correlation times for all states can be found in *Supplementary file 6*.

## Acknowledgements

This work was funded by National Institutes of Health (NIH) grant GM083059 to ALL, and was supported in part by the National Cancer Institute of the NIH under award number P30CA016086 Cancer Center Support Grant. This study made use of NMRbox: National Center for Biomolecular NMR Data Processing and Analysis, a Biomedical Technology Research Resource (BTRR), which is supported by NIH grant P41GM111135 (NIGMS). The content is solely the responsibility of the authors and does not necessarily represent the official views of the NIH. We thank Dr. Stuart Parnham of the UNC Biomolecular NMR Facility for assistance in NMR data collection and Dr. Ashutosh Tripathy of the UNC Macromolecular Interactions Facility for assistance in ITC data collection. We also thank Dr. Qi Zhang for generously providing a 1D selective $^{13}C$ CEST pulse program as well as a script for analysis of CEST data, which were slightly modified for our use here.

## Additional information

### Funding

| Funder | Grant reference number | Author |
|---|---|---|
| National Institutes of Health | GM083059 | Andrew L Lee |
| National Institutes of Health | P30CA016086 | Andrew L Lee |

The funders had no role in study design, data collection and interpretation, or the decision to submit the work for publication.

### Author contributions

Jeffrey P Bonin, Conceptualization, Data curation, Software, Formal analysis, Methodology, Writing – original draft; Paul J Sapienza, Conceptualization, Data curation, Writing – review and editing; Andrew L Lee, Conceptualization, Funding acquisition, Writing – review and editing

### Author ORCIDs

Jeffrey P Bonin ⓘ http://orcid.org/0000-0002-2138-4440
Paul J Sapienza ⓘ http://orcid.org/0000-0001-5279-3653
Andrew L Lee ⓘ http://orcid.org/0000-0003-2783-1907

### Decision letter and Author response

Decision letter https://doi.org/10.7554/eLife.79915.sa1
Author response https://doi.org/10.7554/eLife.79915.sa2

## Additional files

### Supplementary files

- Appendix 1—figure 2—source data 1. apo Δ25 hTS LV $^{13}C$ MQ CPMG 850 MHz.
- Appendix 1—figure 2—source data 2. apo Δ25 hTS LV $^{13}C$ MQ CPMG 600 MHz.
- Appendix 1—figure 2—source data 3. apo Δ25 hTS LV $^{13}C$ SQ CPMG 850 MHz.
- Appendix 1—figure 3—source data 1. 25 µM Δ25 hTS, dUMP ITC.
- Appendix 1—figure 3—source data 2. 50 µM Δ25 hTS, dUMP ITC.
- Appendix 1—figure 3—source data 3. 100 µM Δ25 hTS, dUMP ITC.
- Appendix 1—figure 3—source data 4. 53 µM hTS, TMP ITC.
- Appendix 1—figure 3—source data 5. 150 µM hTS, TMP ITC.

- Appendix 1—figure 3—source data 6. 216 μM hTS, TMP ITC.
- Source code 1. This file is a MATLAB script which reads in a table of intensities from a CPMG experiment (generated by the RateAnalysis module in NMRViewJ) and calculates the $R_{2,eff}$ at each $\nu_{CPMG}$.
- Source code 2. This file is a MATLAB script which performs a 3-state fit (B↔A↔C model) of a single methyl group using SQ and MQ CPMG and CEST data. The parameters for the slow (A↔B) process are fixed based on the values obtained in a global 2-state fit of the CEST data alone. This script requires a fitting function (**Source code 5**), which is where the calculation of the residuals is carried out.
- Source code 3. This file is a MATLAB script which performs a 3-state fit (B↔A↔C model) of a single methyl group using SQ CPMG and CEST data (used for I237 here). The parameters for the slow (A↔B) process are fixed based on the values obtained in a global 2-state fit of the CEST data alone. This script requires a fitting function (**Source code 6**).
- Source code 4. This file is a MATLAB script which performs a global fit of two geminal methyl groups from the same residue using SQ and MQ CPMG and CEST data (used for I101, L131 here). For one of the methyl groups, a B↔A↔C model is used, while for the other a 2-state model is used. The faster process (A↔C) parameters are shared between the two methyl groups, while the slow process (A↔B) parameters are fixed on the values obtained in a global 2-state fit of the CEST data alone. This script requires a fitting function (**Source code 7**).
- Source code 5. This file is the MATLAB fitting function used by **Source code 2**.
- Source code 6. This file is the MATLAB fitting function used by **Source code 3**.
- Source code 7. This file is the MATLAB fitting function used by **Source code 4**.
- Source code 8. This file is a MATLAB script which performs either a 2-state or a 3-state (B↔A↔C model) fit of a single methyl group using SQ and MQ CPMG and CEST data with no fixed parameters. This script requires a fitting function (**Source code 9 and 10**).
- Source code 9. This file is the MATLAB fitting function used by **Source code 8** (3-state).
- Source code 10. This file is the MATLAB fitting function used by **Source code 8** (2-state).
- Source code 11. This file is a MATLAB script which reads in a table of intensities from a CEST experiment (generated by the RateAnalysis module in NMRViewJ) and calculates the normalized intensities at each spin lock offset.
- Source code 12. This file is a MATLAB script which performs a 2-state global fit of multiple methyl groups using only the CEST data. The parameters obtained by this fit are used to fix the slow process parameters in the 3-state fits of the methyl groups involved in the global motion. This script requires a fitting function (**Source code 13**).
- Source code 13. This file is the MATLAB fitting function used by **Source code 12**.
- Source code 14. This file is a MATLAB script which performs a global fit of ITC data collected at various protein concentrations using a general two-site binding model. This script requires a fitting function (**Source code 15**).
- Source code 15. This file is the MATLAB fitting function used by **Source code 14**.
- Source data 1. Apo hTS $^{15}$N $T_1$ 600 MHz.
- Source data 2. Apo hTS $^{15}$N $T_{1\rho}$ 600 MHz.
- Source data 3. Apo hTS $^{15}$N-$^1$H Heteronuclear NOE 600 MHz.
- Source data 4. Apo hTS $^{15}$N $T_1$ 500 MHz.
- Source data 5. Apo hTS $^{15}$N $T_{1\rho}$ 500 MHz.
- Source data 6. Apo hTS $^{15}$N-$^1$H Heteronuclear NOE 500 MHz.
- Source data 7. hTS-dUMP $^{15}$N $T_1$ 600 MHz.
- Source data 8. hTS-dUMP $^{15}$N $T_{1\rho}$ 600 MHz.
- Source data 9. hTS-dUMP $^{15}$N-$^1$H Heteronuclear NOE 600 MHz.
- Source data 10. hTS-dUMP $^{15}$N $T_1$ 500 MHz.
- Source data 11. hTS-dUMP $^{15}$N $T_{1\rho}$ 500 MHz.
- Source data 12. hTS-TMP $^{15}$N $T_1$ 600 MHz.
- Source data 13. hTS-TMP $^{15}$N $T_{1\rho}$ 600 MHz.
- Source data 14. hTS-TMP $^{15}$N-$^1$H Heteronuclear NOE 600 MHz.

- Source data 15. Apo Δ25 hTS $^{15}$N T$_1$ 600 MHz.
- Source data 16. Apo Δ25 hTS $^{15}$N T$_{1\,\rho}$ 600 MHz.
- Source data 17. Apo Δ25 hTS $^{15}$N-$^1$H Heteronuclear NOE 600 MHz.
- Source data 18. Δ25 hTS-dUMP $^{15}$N T$_1$ 600 MHz.
- Source data 19. Δ25 hTS-dUMP $^{15}$N T$_{1\,\rho}$ 600 MHz.

- Supplementary file 1. Goodness of fit and parameter values for 3-state probes. (a) values from global fit with $p_b \cong 1.3\%$, $k_{ex,ab} = 240\ s^{-1}$ (b) population and exchange rate shown for slower of two processes. Unlike the B↔A↔C model, which gives very similar $p_b$ and $k_{ex,ab}$ for individual fits of these probes, the A↔B↔C model yields heterogeneous values, primarily for the exchange rate (c) $\chi^2_{red}$ is calculated as $\frac{\sum_i \left(\frac{residual_i}{error_i}\right)^2}{num\ data\ points - num\ fit\ param}$

- Supplementary file 2. apo hTS µs-ms motion fit parameter values. Probes involved in the concerted process are given the purple background. If a probe is fit to a 2-state model ($A \leftrightarrow C$), the parameters for the $A \leftrightarrow B$ process are left blank. Cases where a $^1$H $\Delta\omega$ is fixed at 0 are indicated by the '-'. Parameter values are reported as median ±standard deviation of fit values from 200 Monte Carlo simulations of the data. For probes involved in the concerted process, the error in $p_b$ is the sum of the errors from the 2-state fit of the CEST data alone and from the global fit of the CPMG and CEST data with the slow process parameters fixed. Only the $^{13}$C $\Delta\omega$'s for the slow, concerted process have sign information from the CEST data; all other $\Delta\omega$'s should be interpreted only as the magnitude of the chemical shift difference. The background of each parameter value is color-coded based on the magnitude of the error relative to the parameter value, where blue indicates low error, yellow indicates medium error, and red indicates large error. In some cases, particularly for the populations, physical constraints on the parameter value (i.e. population cannot be less than 0) lead to highly skewed distributions of the parameter values in our Monte Carlo simulations. This can lead to nonsensical values when reported in our typical manner, for example the $0 \pm 6\%\ p_b$ given for L279 despite the fact that the population cannot be less than 0. In these cases, we have also listed (5% quantile, mode, 95% quantile) to provide greater insight into the distribution of values seen in the Monte Carlo simulations. For L269, marked with the asterisk, the CEST data was not included in the fit. Probes possessing $R_{ex}$ which weren't analyzed include L67, V164, and V313. Refer to the legend of **Supplementary file 7** for a description of the 'met 1' and 'met 2' labels.

- Supplementary file 3. hTS-dUMP µs-ms motion fit parameter values. All probes are fit to a 2-state model. Color-coding and values in parenthesis are as described in **Supplementary file 2**. L192 met 2, marked by the asterisk, is presumably the same methyl group analyzed in apo hTS given the similarity in the $\Delta\omega$'s, but assignment of this signal is complicated by a large chemical shift perturbation between apo and dUMP bound states.

- Supplementary file 4. apo Δ25 hTS µs-ms motion fit parameter values. All probes are fit to a 2-state model. Color-coding and values in parenthesis are as described in **Supplementary file 2**.

- Supplementary file 5. Acquisition parameters for NMR experiments.

- Supplementary file 6. Rotational correlation times for hTS-bound states. Rotational correlation times for various hTS-bound states determined by $^{15}$N relaxation (**Source data 1–19**).

- Supplementary file 7. apo hTS methyl rotation axis order parameters. For LV methyls, the labels 'met1' and 'met2' are given, where 'met1' has the larger $^{13}$C chemical shift, as stereospecific assignments have not been made. For all other states, the 'met1' and 'met2' labels are given based on chemical shift similarity to the apo state. The only exception is L192, where $\Delta\omega$'s obtained from our dispersion fits guided the assignment.

- Supplementary file 8. hTS-dUMP methyl rotation axis order parameters.

- Supplementary file 9. hTS-TMP methyl rotation axis order parameters.

- Supplementary file 10. Δ25 hTS apo methyl rotation axis order parameters.

- Supplementary file 11. Δ25 hTS-dUMP methyl rotation axis order parameters.

- Transparent reporting form

## Data availability

NMR and ITC data have been submitted to Dryad. Data generated and analyzed in this study are included in the manuscript and supporting files (source data/code files).

The following datasets were generated:

| Author(s) | Year | Dataset title | Dataset URL | Database and Identifier |
|---|---|---|---|---|
| Bonin J, Lee A, Sapienza P | 2022 | Human thymidylate synthase NMR relaxation data | https://dx.doi.org/10.5061/dryad.wh70rxwqk | Dryad Digital Repository, 10.5061/dryad.wh70rxwqk |
| Bonin J, Lee A, Sapienza P | 2022 | Human thymidylate synthase ITC data | https://dx.doi.org/10.5061/dryad.j9kd51cfx | Dryad Digital Repository, 10.5061/dryad.j9kd51cfx |

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

## Appendix 1

## Methods

### Additional information on analysis of CPMG and CEST data

As mentioned in the main text, our in-house script for global analysis of CPMG and CEST data calculates $R_{2,eff}$ (CPMG) and normalized intensity (CEST) values from the exchange parameters using the Bloch-McConnell equations. Matrices describing the evolution of the magnetization are given below for the BAC 3-state model. In the case of the MQ CPMG, separate matrices are used for the double quantum (DQ) and zero quantum (ZQ) components of the MQ coherence:

$$
\text{SQ:} \quad
\begin{pmatrix}
-R_{2,SQ} - k_{ab} - k_{ac} & k_{ba} & k_{ca} \\
k_{ab} & -R_{2,SQ} - k_{ba} + i\Delta\omega_{C,ab} & 0 \\
k_{ac} & 0 & -R_{2,SQ} - k_{ca} + i\Delta\omega_{C,ac}
\end{pmatrix}
$$

$$
\text{ZQ:} \quad
\begin{pmatrix}
-R_{2,MQ} - k_{ab} - k_{ac} & k_{ba} & k_{ca} \\
k_{ab} & -R_{2,MQ} - k_{ba} - i\left(\Delta\omega_{H,ab} - \Delta\omega_{C,ab}\right) & 0 \\
k_{ac} & 0 & -R_{2,MQ} - k_{ca} - i\left(\Delta\omega_{H,ac} - \Delta\omega_{C,ac}\right)
\end{pmatrix}
$$

$$
\text{DQ:} \quad
\begin{pmatrix}
-R_{2,MQ} - k_{ab} - k_{ac} & k_{ba} & k_{ca} \\
k_{ab} & -R_{2,MQ} - k_{ba} - i\left(\Delta\omega_{H,ab} + \Delta\omega_{C,ab}\right) & 0 \\
k_{ac} & 0 & -R_{2,MQ} - k_{ca} - i\left(\Delta\omega_{H,ac} + \Delta\omega_{C,ac}\right)
\end{pmatrix}
$$

where $R_{2,SQ(MQ)}$ is the SQ (MQ) transverse relaxation rate, $k_{xy}$ is the rate of transitioning from state x to state y, and $\Delta\omega_{X,yz}$ is the chemical shift difference between states y and z on nucleus X. For analysis of the CEST data, the following evolution matrix was used:

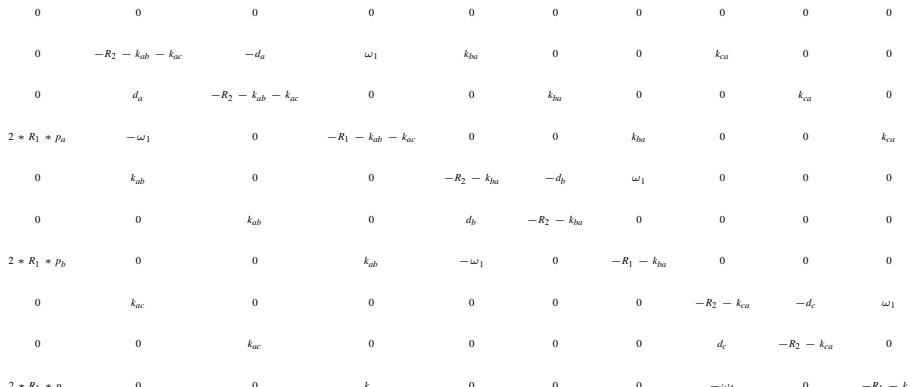

where $R_{1(2)}$ is the longitudinal (transverse) relaxation rate, $p_x$ is the population of state x, $k_{xy}$ is the rate of transitioning from state x to state y, $d_x$ is the chemical shift of state x, and $\omega_1$ is the spin lock power. The chemical shift of the major state, $d_a$, is calculated assuming that the observed resonance is at the population weighted average shift of states a and c (the states connected by the fast processes).

### Loss of slow process in Δ25 hTS shown by 1D selective CEST and CPMG

For $^{13}$C CEST, a 0.2 mM sample of ILV CHD$_2$ apo Δ25 hTS in NMR buffer (99.8% D$_2$O) was prepared as described in the main text (same sample used for $^2$H relaxation). A 1D selective version of the $^{13}$C CEST experiment was run on L192 and L198. Datasets were collected with 25 Hz $^{13}$C spin lock power, 500 Hz power $^2$H decoupling during the relaxation period, and 100 Hz power spin locks ($^1$H and $^{13}$C) for Hartmann Hahn transfers. Peak intensities were obtained in NMRPipe.

For CPMG experiments, a 0.2 mM sample of LV CH$_3$ apo Δ25 hTS in NMR buffer (99.8% D$_2$O) was prepared as described in the main text. MQ CPMG datasets were collected at 850 and 600 MHz. One field (850 MHz) of $^{13}$C SQ CPMG was collected.

## Lack of cooperativity in hTS binding TMP and Δ25 hTS binding dUMP shown by ITC

For ITC experiments with TMP binding to full-length hTS, 0.05, 0.15, and 0.2 mM samples of unlabeled hTS were prepared as described, with the exception that the final buffer used 1 mM TCEP as the reducing agent instead of DTT. ITC experiments were conducted and analyzed as described previously (*Bonin et al., 2019*) using 1.8, 5.2, and 7.7 mM TMP, respectively (*Source code 14 and 15*). For ITC experiments with dUMP binding to Δ25 hTS, 0.025, 0.05, and 0.1 mM samples of unlabeled Δ25 hTS were prepared as described above. The final buffer also included 1 mM EDTA for these samples. ITC experiments used 0.5, 1, and 1.9 mM dUMP, respectively.

## µs-ms motion in folate binding helix shown by $^1$H CPMG 2-plane

A 0.3 mM sample of ILV $CH_3$ apo hTS in NMR buffer (99.8% $D_2O$) was prepared as described in the main text. A $^1$H SQ methyl TROSY CPMG (*Yuwen et al., 2019*) dataset was collected at 850 MHz with 2 planes (50 and 2000 Hz $\nu_{cpmg}$).

## Assessing exposure of fast motion probes using solvent PREs with 3D atom depth index

3D atom depth indices were obtained using the active structure of hTS (PDB 5X5A) and the SADIC (*Alocci et al., 2013*) web server (http://www.sbl.unisi.it/prococoa/). The $C^\beta$ depth index value was used, as we do not have stereospecific assignments of the leucine and valine methyl groups.

## Results

### Comparison of 3-state models for µs-ms motion indicates BAC model is most applicable

8 of the 10 probes involved in the concerted process fit better to a 3-state model than a 2-state model (*Supplementary file 1*). For 3 of these probes – L192, L198, and I237 – the difference in the goodness of fit between the 2 and 3-state (bifurcated) models is quite pronounced ($\left|\Delta\chi^2_{red}\right| =$ 1.49, 0.83, and 2.94, respectively). For the other 5 probes, the difference in goodness of fit is much smaller, though the $\chi^2_{red}$ is smaller for the 3-state fit in all cases. For L101, L131, and L221, while the presence of the slower, concerted process is clear in the CEST data, the faster motion is seen clearly in the other methyl group from the same residue (*Appendix 1—figure 1a*), providing additional justification for applying a 3-state model. For V79 and L121, the slower, concerted process is not clearly seen in the CEST data due to a small $\Delta\omega_{C,ab}$. However, the presence of the slower, concerted process can be seen in the spike in $R_{2,eff}$ at low $\tau_{cp}^{-1}$, which is only captured by the 3-state model (*Appendix 1—figure 1b*).

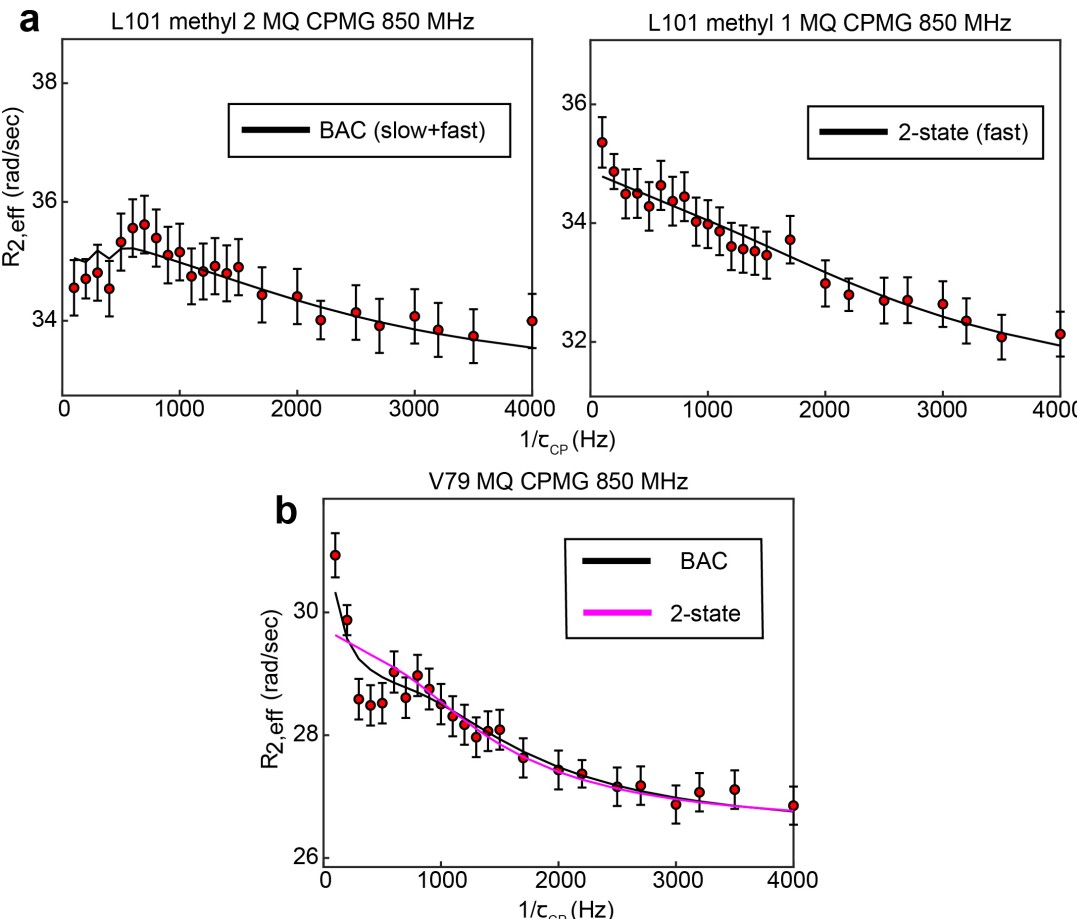

**Appendix 1—figure 1.** B↔A↔C model is most appropriate for many probes involved in the slow, concerted process. 8 of 10 probes involved in the concerted process show 3-state exchange. For some probes, such as L192, L198, and I237, this can be clearly seen by comparison the goodness of fit of the 3-state and 2-state models (***Supplementary file 1***). In other cases, the difference in goodness of fit is much smaller, and additional justification for applying a 3-state model can be gained in other ways. In the case of L101 (methyl 2), while the slow process is clearly seen in the CEST data, the presence of the faster process is supported by the fact that this process is clearly seen in the other methyl group (methyl 1) of the same residue (**a**). In the case of V79, visual inspection of the MQ CPMG profile at 850 MHz shows a clear elevation of the first two points that is captured only by the 3-state fit (**b**). This difference is most likely not reproduced clearly in the $\chi^2_{red}$ because there are many more data points in the CEST data compared to the CPMG, giving the CEST data a larger weight than the CPMG in the norm of the residuals (i.e. the CEST data makes a larger contribution to $\chi^2_{red}$ than the CPMG). Error bars are based on duplicate point RMSD (see Materials & Methods).

For these eight residues, several types of 3-state models are possible. We consider a linear model (A↔B↔C), in which the major state A is connected to a single minor state, and the two minor states are connected to each other, and a bifurcated linear model (B↔A↔C), in which the major state is connected to both minor states. We do not consider a triangular model, as the simpler linear models are sufficient to faithfully reproduce the data. The linear and bifurcated models provide equally good fits ($\left| \Delta\chi^2_{red} \right| <\sim 0.1$) for all of these probes.

For all eight of these 3-state probes, we have chosen to employ the bifurcated model. This decision was based on the populations and exchange rates obtained for the fits to the two different models. With the bifurcated model, all 8 of these probes are well fit with $k_{ex,ab} = 240$ s⁻¹ and $p_b \cong$ 1.3% ($p_b$ can vary slightly between these probes to account for variability in $p_c$, the individual rates $k_{ab}$ and $k_{ba}$ are identical however), indicating that they are involved in a concerted process. In the case of the linear model, however, the populations and exchange rates are not in agreement between these probes (***Supplementary file 1***). Thus, employing the linear model would be to say that the agreement in population and exchange rate seen in the case of the bifurcated model is

purely coincidental, which is unlikely. Consequently, we prefer the bifurcated model, which has been used throughout the main text.

## Δ25 hTS lacks the slower, concerted process, yet still displays the faster µs-ms motion

1D selective CEST data were collected on L192 and L198, in addition to 3 CPMG datasets (MQ 850 and 600 MHz, $^{13}$C SQ 850 MHz), in apo hTS Δ25 (*Appendix 1—figure 2—source data 1–3*). The CPMG data are fit well by a 2-state model for all probes, and the probes showing 3-state exchange involving the slow, concerted process in the full-length apo enzyme have parameter values similar to those of the faster process from the full length (compare V79, L121, L192, and L198 in *Supplementary files 2 and 4*, *Appendix 1—figure 2b*). Consistent with this, the CEST data on L192 and L198 of Δ25 hTS lack a minor state dip (*Appendix 1—figure 2c*), further demonstrating the loss of the slow, concerted process. Notably, the parameter values do not match the full length fast process values for all probes. Consider L259, which has $k_{ex,ac}$ = 860 ± 70 s$^{-1}$ and |$\Delta\omega_{C,ac}$| = 2.7 ± 0.2 ppm in the full length, but $k_{ex}$ = 3800 ± 200 s$^{-1}$ and |$\Delta\omega_C$| = 1.5 ± 0.2 ppm in the truncation. This indicates that the N-terminus influences these motions, at least for some probes. Overall, the agreement between $^{13}$C |$\Delta\omega$| in full length (faster process) and Δ25 is reasonably good (*Appendix 1—figure 2b*). In addition, the ground state chemical shifts are virtually identical for the full length and truncated enzymes (*Figure 7* of the main text). Together, these results suggest that similar configurations are being accessed in both cases.

## hTS binding TMP and Δ25 hTS binding dUMP lack cooperativity

ITC experiments were conducted on hTS binding TMP and Δ25 hTS binding dUMP (*Appendix 1—figure 3—source data 1–6*) as we have done previously for full-length hTS binding dUMP (*Bonin et al., 2019*). Fits to these data using a general two-site binding model show that in both of these cases, unlike with full length hTS binding dUMP, the free energies of the two binding events are nearly equivalent, indicating a lack of cooperativity (*Appendix 1—figure 3*). Further, we find that the individual binding enthalpies and entropies are nearly equivalent for the two binding events in each of these cases. Interestingly, the binding enthalpies and entropies for Δ25 hTS binding dUMP are both similar to the second binding enthalpy and entropy of full length hTS binding dUMP, consistent with the idea that there is some process coupled to the first binding event in full-length hTS that is absent in Δ25 hTS, as we discuss in the main text (interactions of N-terminus with the protein surface). In addition, we note that the binding enthalpies and entropies for full length hTS binding TMP are similar to the first binding enthalpy and entropy of full-length hTS binding dUMP; however, because TMP has the additional methyl group—and can thus make additional interactions with the protein—the interpretation is less straightforward in this case.

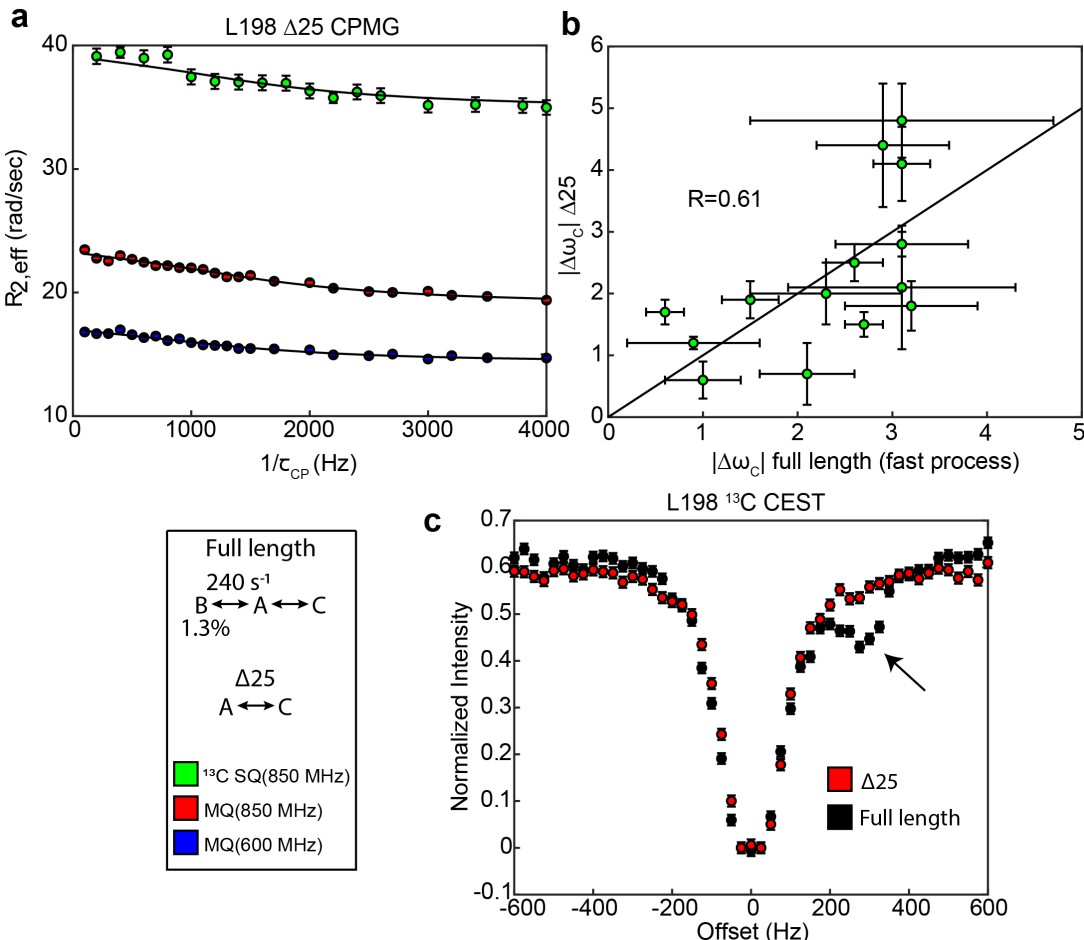

**Appendix 1—figure 2.** Δ25 hTS lacks the slow, concerted process, but retains faster µs-ms motion. Global fits of 3 CPMG datasets (2 fields of MQ, 1 field of $^{13}$C SQ, *Appendix 1—figure 2—source data 1–3*) for individual probes show that the probes undergoing 3-state exchange in full length apo hTS are well described by a 2-state model in apo Δ25 hTS (**a**). Error bars are based on duplicated point RMSD (see Materials & Methods). $^{13}$C Δω's for Δ25 hTS agree reasonably well with those from the fast process for full length apo hTS (**b**), suggesting that similar configurations are being accessed in both cases. Error bars are based on fits to 200 Monte Carlo simulated datasets (see Materials & Methods). 1D selective CEST on L198 confirms the loss of the slow process in Δ25 hTS (**c**). Error bars are based on 1x base plane noise.

The online version of this article includes the following source data for appendix 1—figure 2:

**Appendix 1—figure 2—source data 1.** apo Δ25 hTS LV $^{13}$C MQ CPMG 850 MHz.

**Appendix 1—figure 2—source data 2.** apo Δ25 hTS LV $^{13}$C MQ CPMG 600 MHz.

**Appendix 1—figure 2—source data 3.** apo Δ25 hTS LV $^{13}$C SQ CPMG 850 MHz.

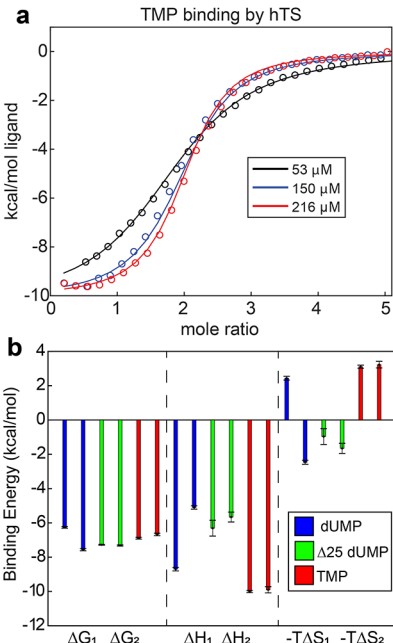

**Appendix 1—figure 3.** Lack of cooperativity in full length hTS binding TMP and Δ25 hTS binding dUMP. (**a**) Global fit of ITC data for TMP binding to hTS with three enzyme concentrations (**Appendix 1—figure 3—source data 1–3**). (**b**) $\Delta G_i$, $\Delta H_i$, and $-T\Delta S_i$ for binding event $i$ from ITC fits are shown for full length hTS binding dUMP (blue), Δ25 hTS binding dUMP (green, **Appendix 1—figure 3—source data 4–6**), and full length hTS binding TMP (red, **Appendix 1—figure 3—source data 1–3**). $\Delta G_2$ is ~1.3 kcal/mol more negative than $\Delta G_1$ for full length hTS binding dUMP, indicating ~9 fold positive cooperativity (**Bonin et al., 2019**). For both Δ25 hTS binding dUMP and full length hTS binding TMP, $\Delta G_1 \cong \Delta G_2$, indicating that the binding is non-cooperative. Error bars are based on fits to Monte Carlo simulated datasets (see Materials & Methods).

The online version of this article includes the following source data for appendix 1—figure 3:

**Appendix 1—figure 3—source data 1.** 25 µM Δ25 hTS, dUMP ITC.

**Appendix 1—figure 3—source data 2.** 50 µM Δ25 hTS, dUMP ITC.

**Appendix 1—figure 3—source data 3.** 100 µM Δ25 hTS, dUMP ITC.

**Appendix 1—figure 3—source data 4.** 53 µM hTS, TMP ITC.

**Appendix 1—figure 3—source data 5.** 150 µM hTS, TMP ITC.

**Appendix 1—figure 3—source data 6.** 216 µM hTS, TMP ITC.

