## [Editor Report]

Bonin et al. provide important new insights into the nucleotide substrate binding cooperativity of homodimeric human thymidylate synthase (hTS). hTS converts deoxyuridine monophosphate (dUMP) into deoxythymidine monophosphate (dTMP) and is therefore a target for cancer therapies. Extensive use of methyl-based NMR spectroscopy provides a set of convincing data that support a number of new insights into the conformational preferences of the apo state and the role of conformational entropy in mediating cooperativity within this enzyme. In addition, the findings point to a key role for the intrinsically disordered N-terminal region of hTS in the dynamic changes that occur upon binding of the dUMP substrate. The work very nicely demonstrates the power of NMR spectroscopy in elucidating ensemble conformations and dynamics and creates new ways to think about targeting this essential enzyme.

---

## [Decision Letter]

**Decision letter after peer review:**

Thank you for submitting your article "Dynamic allostery in substrate binding by human thymidylate synthase" for consideration by *eLife*. Your article has been reviewed by 3 peer reviewers, and the evaluation has been overseen by a Reviewing Editor and Volker Dötsch as the Senior Editor. The following individual involved in the review of your submission has agreed to reveal their identity: Vincenzo Venditti (Reviewer #1).

Essential revisions:

Each of the three reviewers provided detailed feedback for strengthening and clarifying the manuscript. Please address the specific points listed under "Recommendations for the authors" by Reviewers 1 and 3. Reviewer 2 also provides specific comments in the "Recommendations for the authors" section but includes requests for revisions and clarifications throughout the "Public review" section. Please be sure to address all issues raised by the reviewers with a focus on addressing the first comment by Reviewer 2: "Generally, needs better descriptions of the assumptions made, and their limitations."

*Reviewer #1 (Recommendations for the authors):*

I have only a few suggestions to improve the quality of this work:

1) The authors measured solvent-PREs and compared these solvent accessibility data with the depth of the NMR-active probes in the crystal structure of hTS. It was shown that the 3D atom depth index is a much better parameter to compare to experimental solvent-PREs than 1D atom distances from the molecular surface (see https://doi.org/10.1093/bioinformatics/bti444). In my opinion, the authors should confirm that the discrepancies between experimental PREs and surface accessibility persist even when the atom depth index is used to estimate the atomic depth within the structure. A server to calculate the atom depth indexes is at http://sadic.sourceforge.net.

2) The authors note the presence of active site residues that exhibit higher solvent accessibility than would be expected based on the crystal structure of hTS. The authors should acknowledge that this is a known feature, common to several enzymes. See https://doi.org/10.1016/j.pnmrs.2008.10.003 for a review on this topic.

3) The RDC analysis was performed using ARTSY spectra. This implies that the backbone resonance assignment of hTS is known. Can the authors report the secondary Calpha and Cbeta chemical shifts for the apo and holoenzyme and see if there is any indication that ligand binding induces structuring of the N-terminal tail?

*Reviewer #2 (Recommendations for the authors):*

1. Generally, needs better descriptions of the assumptions made, and their limitations.

2. Describe structural features of the active and inactive conformations. Why are they called that? What differences would be expected to be detectable by NMR. If the only difference is in a loop (as described on page 3), why are the predicted RDCs so different? (Figure 4) Why should "inactive" bind more weakly to dUMP?

3. Include representative RDC spectra; explain why unaligned apo is a good reference for aligned bound.

4. Not clear if KDEs of ΔS^2 are weighted by uncertainty; no y-axes? Why not plot a histogram or by-residue values? KDE curves for FL and Δ25 are very similar; do we care which residues become less dynamic?

5. Figure 8 is not a true thermodynamic box, as energy is a state function so TdS must be independent of path. This should be reformulated.

*Reviewer #3 (Recommendations for the authors):*

Page 9 – The description of the CEST/CPMG fitting is a bit unclear. Were probes undergoing the same exchange process, globally fitted?

For the 3-state model fits, rate constants were fitted instead of exchange rates and populations because "only the rates (and not the populations) are identical amongst the probes". But the ratio of the rates gives the population and the sum of the rates give kex, so there is no difference?

Also, it is unclear why geminal methyl groups should be fit globally

Page 17, lines 12-15 – These probes are clearly in a fast exchange where kex and δ omega will be correlated (even though the data was collected at 2 fields the 850 MHz data might not be enough to break the correlation). The large values in δ omega might not be that large.

Page 18, line 22, and page 19 lines 1-2 – It is unclear why the observed enhancement in solvent exposure is consistent with the large 13C δ omegas from the CPMG. This point needs to be clarified.

Page 20, lines 5-17 – I think the organization would flow better if this paragraph was before the solvent PREs. Since the CPMG/CEST analysis in the presence of substrate strongly supports the model that the slow process is related to movement between active/inactive conformations. This then flows into solvent PREs to address the faster exchange in both the apo and dUMP-bound states. It might also help to move the RDC section to before the solvent PREs, as these also address the main point of the paper.

---

## [Author Response]

Essential revisions:Reviewer #1 (Recommendations for the authors):I have only a few suggestions to improve the quality of this work:1) The authors measured solvent-PREs and compared these solvent accessibility data with the depth of the NMR-active probes in the crystal structure of hTS. It was shown that the 3D atom depth index is a much better parameter to compare to experimental solvent-PREs than 1D atom distances from the molecular surface (see https://doi.org/10.1093/bioinformatics/bti444). In my opinion, the authors should confirm that the discrepancies between experimental PREs and surface accessibility persist even when the atom depth index is used to estimate the atomic depth within the structure. A server to calculate the atom depth indexes is at http://sadic.sourceforge.net.

We have used SADIC to calculate 3D atom depth indices for the ILV residues in the hTS active structure, and an analysis of the sPRE data using these 3D depth indices rather than the 1D depths is presented in Figure 3—figure supplement 1. Overall, we find that the result is very similar using the 3D atom depth indices, which once again highlight the active site and extended loop as being more heavily influenced by the paramagnetic cosolute than one would expect given their depths. We have opted to use the 1D atom depths in the main text figure, as we feel that the theoretical dependence of the PRE on the 1D distance aids in the interpretation of these data.

2) The authors note the presence of active site residues that exhibit higher solvent accessibility than would be expected based on the crystal structure of hTS. The authors should acknowledge that this is a known feature, common to several enzymes. See https://doi.org/10.1016/j.pnmrs.2008.10.003 for a review on this topic.

We thank Reviewer 1 for bringing this to our attention. We have added text on page 12 noting that this same observation has been made in other enzymes.

3) The RDC analysis was performed using ARTSY spectra. This implies that the backbone resonance assignment of hTS is known. Can the authors report the secondary Calpha and Cbeta chemical shifts for the apo and holoenzyme and see if there is any indication that ligand binding induces structuring of the N-terminal tail?

Indeed, we have backbone assignments for many residues in the N-terminus. Figure 8—figure supplement 1 has been modified to include a plot of the change in C^α^ chemical shift between apo and dUMP bound forms of hTS, showing essentially no change in these shifts for N-terminal probes. In addition, text has been added on page 28 discussing this result. While the C^β^ shifts are not shown, we also see very little change there upon dUMP binding. This further strengthens the argument that there is no noticeable structuring of the N-terminus upon dUMP binding.

Reviewer #2 (Recommendations for the authors):1. Generally, needs better descriptions of the assumptions made, and their limitations.

Text has been added on pages 12 and 15 explicitly noting key assumptions that are made in the work.

2. Describe structural features of the active and inactive conformations. Why are they called that? What differences would be expected to be detectable by NMR. If the only difference is in a loop (as described on page 3), why are the predicted RDCs so different? (Figure 4) Why should "inactive" bind more weakly to dUMP?

The manuscript indeed had minimal description of these two conformations. Text has been added on pages 7 and 8 discussing the differences between the active and inactive conformations, why they are named as such, and why one would expect the inactive conformation to have a significantly weakened affinity for dUMP. As is now described on pages 7 and 8, there are differences between the two beyond the active site loop (these include the extended loop as well as smaller displacements of the backbone throughout the molecule), which is why there are differences in the predicted RDCs.

3. Include representative RDC spectra; explain why unaligned apo is a good reference for aligned bound.

One would expect the one bond scalar couplings between the proton and nitrogen of the backbone amides to be very similar between the apo and substrate bound forms. Further, any minor differences in the scalar couplings would be negligible compared to modulation of observed couplings resulting from partial alignment by phage. The agreement observed between the dUMP-bound experimental and predicted RDCs supports the validity of this expectation.

4. Not clear if KDEs of ΔS^2 are weighted by uncertainty; no y-axes? Why not plot a histogram or by-residue values? KDE curves for FL and Δ25 are very similar; do we care which residues become less dynamic?

We prefer the KDE plots over typical histograms as we found them to be less cluttered and more aesthetically appealing, though they convey the same information. The purpose of the KDE plots is to show the distribution of ΔSaxis2 values, and in particular to give a feel for the average change in order parameter for each case, which is what the entropy meter relates to the change in conformational entropy. As a result, the residues associated with each of those individual values is not relevant. However, Figure 5b shows the changes in order parameter plotted on the structure. The KDE plots for full length and Δ25 are indeed quite similar, though there is clearly an overall reduction in ΔSaxis2 values in the truncation; the entropy meter indicates that for a protein the size of hTS, even these fairly small differences in the average change in order parameter are sufficient to produce noticeable differences in the changes in conformational entropy.

5. Figure 8 is not a true thermodynamic box, as energy is a state function so TdS must be independent of path. This should be reformulated.

This is an interesting point. The values along paths A and B do not yield exactly identical results because the collection of probes used to calculate those values are not identical. If identical sets of probes are used to calculate the changes at each step, the values obtained along paths A and B are indeed exactly identical. We chose to report the data in this way because we wanted to use all of the available data, and we were pleased that including the “extra” data didn’t qualitatively change the result (the changes in conformational entropy obtained along paths A and B are still well within error of each other). Text has been added to the legend of Figure 8 to clarify this point.

Reviewer #3 (Recommendations for the authors):Page 9 – The description of the CEST/CPMG fitting is a bit unclear. Were probes undergoing the same exchange process, globally fitted?

We agree that the description of this uncommon fitting approach could be improved, and we thank the reviewer for drawing attention to this. For the probes involved in the concerted process, a global, 2-state fit was carried out with the CEST data to determine the exchange parameters for this process. For the subset of these probes which showed 3-state behavior (i.e. 2-state fits were not able to reproduce the CPMG data), local 3-state fits were carried out using both the CPMG and CEST data to determine the exchange parameters for the faster (non-concerted) motions. For these 3-state fits, the parameters for the concerted process were fixed using their values from the global CEST fit. We find that performing a “true” global fit of these probes using both the CPMG and CEST data gives the same answer, but takes considerably more time to run than the approach we used. Text has been added to page 7 and page 34 to attempt to clarify this.

For the 3-state model fits, rate constants were fitted instead of exchange rates and populations because "only the rates (and not the populations) are identical amongst the probes". But the ratio of the rates gives the population and the sum of the rates give kex, so there is no difference?

The ratio of the rates gives the *relative value* of the populations, i.e. kabkba=pbpa. This is important in our case because for the 3-state probes, pc is variable (the faster process is not concerted). Since the populations of states A, B, and C must sum to 1, this means that as pc increases (decreases), the populations of states A and B must decrease (increase) accordingly. Thus, while kab, kba, and pbpa are identical for all the probes involved in the concerted process, pb and pa themselves are not the same. After the individual rates have been fit, the populations and kex ’s can be calculated from those rates as you described.

Also, it is unclear why geminal methyl groups should be fit globally?

Given that these methyl groups are part of the same side chain, one might expect even a fairly localized motion to involve both of them. Of course, it is not a given that this will be true, but in our case, we found that all of the geminal methyl pairs fit well globally.

Page 17, lines 12-15 – These probes are clearly in a fast exchange where kex and δ omega will be correlated (even though the data was collected at 2 fields the 850 MHz data might not be enough to break the correlation). The large values in δ omega might not be that large.

It is certainly true that the errors in these Δω’s are quite large (as can be seen in Figure 2d). However, even at the low end of their error bar, the values of most of these Δω’s would still be at least as large as the largest Δω seen in the slow process (1.4 ppm). Text has been added to page 9 echoing this response.

Page 18, line 22, and page 19 lines 1-2 – It is unclear why the observed enhancement in solvent exposure is consistent with the large 13C δ omegas from the CPMG. This point needs to be clarified.

The rationale is that exposing these buried probes would involve a relatively large perturbation to the structure, and thus would give rise to a large change in chemical shift. Text has been added to page 12 along these lines.

Page 20, lines 5-17 – I think the organization would flow better if this paragraph was before the solvent PREs. Since the CPMG/CEST analysis in the presence of substrate strongly supports the model that the slow process is related to movement between active/inactive conformations. This then flows into solvent PREs to address the faster exchange in both the apo and dUMP-bound states. It might also help to move the RDC section to before the solvent PREs, as these also address the main point of the paper.

We agree that moving the sPRE section improves the flow of the paper, and thank the reviewer for this suggestion. The sPRE section has been moved to after the discussion of the CPMG analysis of dUMP-bound hTS.